# ASSEMBLY-R1: 3D ASSEMBLY REASONING VIA RL-BASED VISION LANGUAGE MODELS

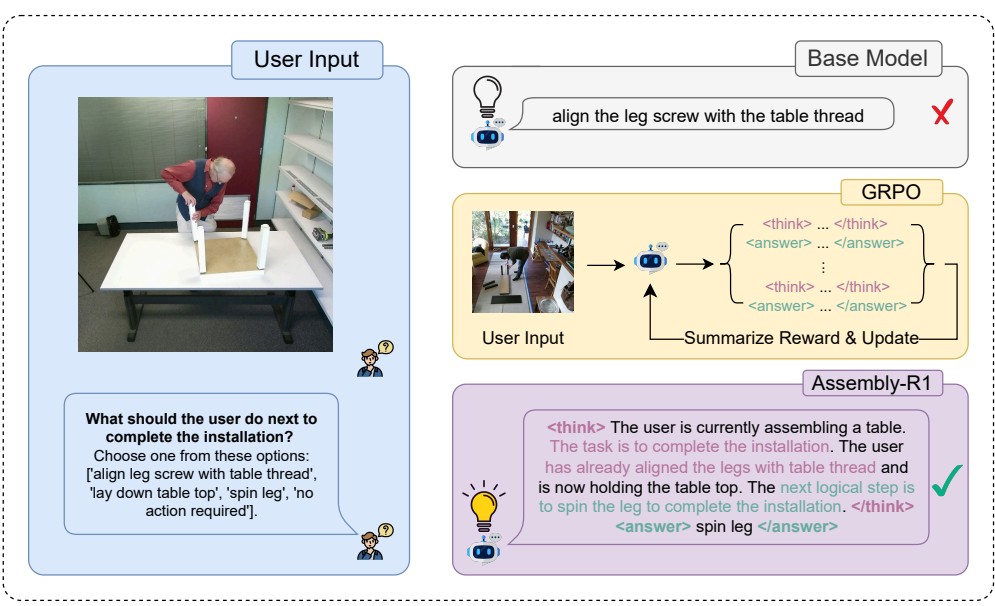

Figure 1: **Bridging the gap in Part Assembly Planning.** We propose Assembly-R1, a model trained to reason about spatial relationships via Reinforcement Learning. The diagram illustrates how our GRPO-based training pipeline (middle) transforms a failing base model (top) into an agent capable of precise state analysis and correct next-step planning (bottom), using a representative example from our FurniQA.This model shows great potential for home-use and industrial intelligent robots.

## ABSTRACT

Part assembly requires agents to possess precise spatial interpretation and multi-step structural reasoning. While large Vision-Language Models (VLMs) have shown promising capabilities in general Visual Question Answering (VQA), existing benchmarks inadequately reflect the complexities inherent in assembly reasoning. To bridge this gap, we introduce FurniBench, an assembly-specific VQA benchmark, coupled with FurniQA, a large-scale dataset targeting part recognition, connectivity reasoning, step planning, etc. Using Qwen2-VL-2B-Instruct as a base model, with 39.1% accuracy on FurniBench, we first establish a supervised fine-tuning (SFT) baseline, which highlights both the benefits and the limitations of SFT in this domain. Building on this, we propose Assembly-R1, a model trained via Group Relative Policy Optimization (GRPO). With enhanced reasoning capabilities, Assembly-R1 acheives an accuracy of 73.6%, outperforming other baselines on FurniBench by a large margin. Furthermore, the consistent gain on zero-shot performance on Out-of-Domain (OOD) spatial understanding and Embodied AI benchmarks indicates that Assembly-R1 acquires transferable spatial skills applicable to broader Embodied AI scenarios. This work establishes FurniBench as a critical resource for both diagnosing deficits in current VLMs and teaching the fundamental structural reasoning required for down-stream applications. We will release the dataset and code upon acceptance of this paper.

# 1 INTRODUCTION

Imagine a scenario where a user attempts to assemble a piece of furniture. Following a plain printed installation guide can be confusing, especially when instructions are incomplete or ambiguous. Similarly, in industrial automatic assembly, operators often face the challenge of interpreting complex 3D assembly environments under tight constraints. In both cases, intelligent robotic systems capable of understanding visual scenes and responding to natural language queries in a linguistic manner would significantly assist humans in completing the task. This capability falls within the scope of Visual Question Answering (VQA), a popular research area at the intersection of Computer Vision (CV) and Natural Language Processing (NLP) that was introduced by Agrawal et al. Antol et al. (2015).

Over the past years, VQA has evolved from answering simple, closed-form questions to addressing more complex reasoning and abstract challenges Pandey et al. (2025). Researchers have extended VQA to various application domains, such as medical imaging Bazi et al. (2023); He et al. (2020); Al-Hadhrami et al. (2023), robotics Firoozi et al. (2023); Jiang et al. (2023); Shirai et al. (2024), Embodied AI Li et al. (2023b); Ma et al. (2023); Lee et al. (2022), education and training Huynh et al. (2025); Pandey et al. (2025), etc. The applications of large vision-language models (VLMs) have further enhanced the capabilities of VQA systems by enabling deeper visual-textual alignment and contextual understanding. Brown et al. describe language models as "few-shot learners" Brown et al. (2020), indicating their potential for generalization across diverse tasks with minimal supervision. In this context, VLMs have demonstrated their promising performance on a range of VQA benchmarks Alayrac et al. (2022); Li et al. (2022); Qi et al. (2024); Li et al. (2023a).

Despite these advances, there remains a gap in applying VQA and VLM to the specific domain of 3D part assembly tasks. These tasks are challenging because they involve a mixture of closed vocabulary problems (e.g. part recognition) and open vocabulary questions (e.g. spatial understanding), where answer spaces may vary by context Eichstaedt et al. (2021); Wu et al. (2024); Ko et al. (2023). Keeping the alignment between different modalities is a prerequisite for accomplishing these tasks. In addition, these tasks require the model's deeper understanding of the scenario, such as spatial relationships, reasoning about physical constraints among components and the environment, and interpreting ambiguous human instructions in context-dependent scenarios Yan et al. (2020); Suárez-Ruiz & Pham (2015); Jia et al. (2025); Zhan et al. (2020); Cheng et al. (2023); Zhang et al. (2022). These demands go beyond what general-purpose VLMs are typically designed to handle.

To fill this gap, we introduce FurniBench, a benchmark specifically tailored for part-assembly tasks. It comprises 3 main categories of assembly-related queries and 15 subcategories, designed to capture the diverse challenges of visual question answering in this domain. Alongside the benchmark, we present FurniQA, a dataset constructed for FurniBench, containing around 1.6 million high-quality QA pairs derived from the IKEA ASM Dataset Ben-Shabat et al. (2021), providing a new platform for researchers to investigate assembly-related VQAs under real-world scenarios. Given the limitations of existing VLMs in handling such domain-specific tasks, we adopt Qwen2-VL-2B-Instruct as the base model Wang et al. (2024) and first establish a supervised fine-tuning (SFT) baseline using 15k randomly sampled QA pairs from FurniQA. While SFT provides initial performance gains, it also exposes issues such as task-specific overfitting and reduced generalization. To address these challenges, and inspired by the reasoning-enhancement framework of DeepSeek-R1 Guo et al. (2025), we employ Group Relative Policy Optimization (GRPO) Shao et al. (2024) with multi-granular rewards to foster self-reflective reasoning capabilities. This Reinforcement Learning (RL) approach equips the model with stronger Chain-of-Thought (CoT) reasoning, leading to more accurate and generalizable performance.

We utilize answer accuracy as the primary metric to evaluate model performance. Our SFT baseline, Assembly-V1, achieves 69.6% on FurniBench, while our reasoning model, Assembly-R1, further improves performance to 73.6%. Both models significantly outperform the base model, Qwen-2-VL-2B-Instruct, which attains only 39.1%, and even large-scale closed-source VLMs, like GPT-4o OpenAI (2024a) and Gemini-2.5-Pro Google DeepMind (2025) on FurniBench. These results underscore the value of task-specific fine-tuning while demonstrating that RL-based optimization yields substantial additional gains without requiring extra annotated data. Furthermore, we evaluate zero-shot generalizability across multiple Out-of-Domain (OOD) benchmarks Lee et al. (2022); Tong et al. (2024); Hudson & Manning (2019); Ma et al. (2023); Li et al. (2023b). While Assembly-

V1 suffers from catastrophic forgetting, Assembly-R1 demonstrates consistent gains, empirically reinforcing the "SFT Memorizes, RL Generalizes" hypothesis proposed by Chu et al. (2025).

**Contributions:**

- We propose a new benchmark called FurniBench, designed for Visual Question Answering (VQA) in part assembly scenarios, aiming to evaluate models' 3D spatial reasoning and step planning abilities.
- We introduce FurniQA, a large-scale dataset comprising 1.6M diverse assembly-related visual QA pairs, spanning 3 major question categories and 15 specific task types. Derived from the IKEA ASM Dataset, FurniQA is tailored for assembly-focused VQA and, with embedded frame IDs, can be readily extended to assembly-related VideoQA tasks.
- We establish a supervised fine-tuning (SFT) baseline, Assembly-V1, based on Qwen2-VL-2B-Instruct, which demonstrates notable improvements over the base model (69.6% vs. 39.1%), while also highlighting the limitations of SFT in robustness and generalization.
- We are the first to apply Group Relative Policy Optimization (GRPO) for reasoning enhancement in VLMs targeting 3D structural understanding. The reasoning model, Assembly-R1, achieves 73.6% accuracy, outperforming both the base model and the SFT baseline, while requiring no additional annotated supervision. It also achieves promising OOD performance, indicating its generalizability to more downstream tasks.

## 2 RELATED WORKS

### 2.1 VISION LANGUAGE MODEL AND VISUAL QUESTION ANSWERING

Recent advancements in Vision-Language Models (VLMs) have significantly improved multimodal understanding. Models such as Flamingo, BLIP, and BLIP-2 Alayrac et al. (2022); Li et al. (2022; 2023a) have demonstrated impressive performance by effectively aligning visual and textual modality. OpenAI GPT-4o OpenAI (2024a) marks a major milestone in multimodal integration, achieving state-of-the-art in various benchmarks. Meanwhile, the emergence of open-source VLMs, like QwenVL, InternVL, LLaVA, etc. Bai et al. (2023); Wang et al. (2024); Bai et al. (2025); Chen et al. (2024b); Zhu et al. (2025); Liu et al. (2023) has largely boosted the research in Visual Question Answering (VQA). At the same time, researchers have developed a variety of benchmarks to evaluate models and explore their full potential in multiple aspects Singh et al. (2019); Schwenk et al. (2022); Tong et al. (2024); Ma et al. (2023). The co-evolution of VQA benchmarks and VLMs continuously pushes forward the development of more robust and capable models.

### 2.2 MODEL REASONING WITH REINFORCEMENT LEARNING

Following the success of large language models (LLMs) in general knowledge tasks Touvron et al. (2023); Radford et al. (2018); Brown et al. (2020), researchers have increasingly turned their attention to enhancing models' reasoning abilities, particularly for more complex domains such as science, mathematics, and logic OpenAI (2024b); Guo et al. (2025). OpenAI o1 model demonstrates that incorporating Reinforcement Learning (RL) allows models to learn from feedback on their generated responses, leading to Chain-of-Thought (CoT) reasoning patterns and more accurate answers. DeepSeek introduces R1-Zero Guo et al. (2025), a GRPO model Shao et al. (2024) to improve reasoning ability without relying on additional supervised data. With a simple rule-based reward design, it achieves competitive performance on reasoning benchmarks at only a fraction of the training cost compared to its counterparts, largely reducing the training requirement for hardware.

In the vision-language domain, SpatialVLM addresses the limitations of existing vision-language models in spatial reasoning by training on an Internet-scale multimodal dataset rich in spatial relationships Chen et al. (2024a). Inspired by DeepSeek-R1, VLM-R1 and VisualThinker-R1-Zero reproduce the 'aha' moment with non-SFT GRPO method on various VQA benchmarks Shen et al. (2025); Zhou et al. (2025).

Overall, these works demonstrate the growing impact and potential of using reinforcement learning-based methods to enhance the existing base model's reasoning capabilities with reduced reliance on annotated data and training resources.

### 2.3 IKEA ASM DATASET

The IKEA ASM dataset is a richly annotated, multimodal, and multiview video dataset of furniture assembly tasks Ben-Shabat et al. (2021). Originally designed for benchmarking tasks such as video action recognition, object segmentation, part tracking, and human pose estimation, it comprises 371 video samples, including 48 unique assemblers constructing four different types of IKEA furniture in five distinct environments. Every video includes recordings from three camera views, and the primary view (denoted as 'top') contains an RGB-D stream, atomic action labels, human pose estimation, object and part tracking, etc.

## 3 METHOD

### 3.1 PROBLEM FORMULATION - FURNIBENCH

FurniBench is a VQA benchmark designed for assessing models' performance on assembly-related tasks. Given a visual input $v$ and a textual question $q$, the task is to predict an answer $o$ that matches the reference answer $o_{ref}$. Formally, the model learns a function: $f_\theta : (v, q) \rightarrow o$, where $\theta$ denotes the trainable parameters, optimized to minimize the discrepancy between $o$ and $o_{ref}$.

### 3.2 DATASETS - FURNIQA

**First Assemble Pair**

**Q:** Which two parts can be assembled first?
**A:** (K, H), (V, H), (J, H), (E, H)

**Single Part Recognition**

**Q:** What is the part labeled in N?
**A:** Table Shelf

**Action Recognition**

**Q:** What is the user doing in this frame?
**A:** Align leg screw with table thread

**Next Step Inference**

**Q:** What should the user do next to complete the installation?
**A:** Spin table leg

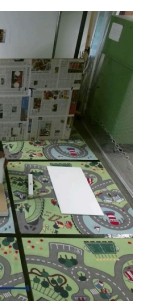
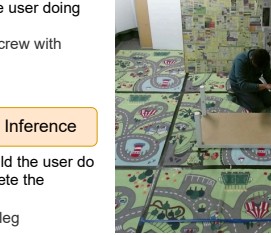

**Object Recognition**

**Q:** What is the most likely furniture type shown by these parts?
**A:** Shelf Drawer

**Installation Preparation**

**Q:** Is any additional step required before installing the side panel?
**A:** Align side panel holes with front panel dowels

**Part Set Completeness**

**Q:** Are all the detachable parts been labeled correctly?
**A:** No

**First Dissemble Part**

**Q:** Which part(s) can be dissembled first?
**A:** C

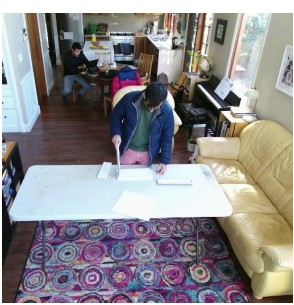
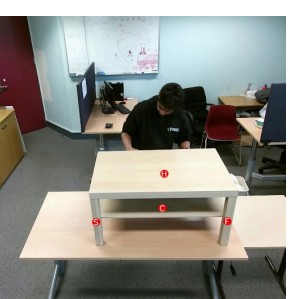

Figure 2: A demonstration of example QA pairs from FurniQA. Visual inputs are shown at the center, surrounded by corresponding textual questions and reference answers. Different QA task categories are highlighted in distinct colors, reflecting the diversity of research challenges covered in FurniBench.

To build our dataset, FurniQA, in the context of 3D assembly understanding, we utilize the RGB stream from the main camera view of the IKEA ASM video streams and combine each visual frame with its corresponding annotations. All QA pairs in FurniQA are programmatically generated using predefined rules grounded in the dataset's annotations. Importantly, the questions are manually calibrated by humans to ensure they are reasonable and aligned with real-world assembly scenarios. No generative models were used in answer generation, ensuring the validity and reliability of each QA pair.

Based on the stage of the assembly process, each scene is categorized into one of three phases: Beginning, In Progress, or Completed. QA pairs are tailored according to these phases to ensure that the questions are contextually relevant and reflective of real-world scenarios. FurniQA comprises approximately 1.6 million QA pairs and is organized into three main task categories: Part Recognition, Part Connectivity, and General Assembly Understanding. The specific task types and corresponding quantities are detailed in the Appendix. The objectives of each main category are as follows:

**Part Recognition** challenges the model's capability in identifying individual furniture parts, like drawer side panels or the table shelf, understanding the part set completeness by assessing whether all required parts are present in the scene, and inferring the identity of the final object (e.g. a table or a bench) based on dispersed parts.

**Part Connectivity** requires the model to understand the topological and physical relationships among parts. For example, it should determine which parts can be assembled, and in what sequence. Some tasks even include reverse reasoning, such as identifying which part can be disassembled first, pushing the model to demonstrate a deeper understanding of the structural dependencies and assembly logic.

**General Assembly Understanding** is designed around the atomic actions of the assembly process. The model is expected to recognize the current or infer the next assembly steps, e.g., picking up a specific part or aligning two components. These questions are specifically challenging as they require the model to: (1) comprehend the current state of the scene, including parts already assembled and those remaining; (2) reason about correct assembly sequence, like which steps should be performed first ahead of a specific step; (3) differentiate between preparatory (e.g. pick up or align parts) and active assembly actions (e.g. insert or attach parts).

### 3.2.1 REDUCING BIAS AND SUBJECTIVITY

- **Increasing Diversity** Questions are rephrased with GPT-4o to increase the diversity of expressions. The question expression variations are listed in the Appendix.

- **Avoiding Enforced Single Answer** IKEA-ASM includes multiple assembly demonstrations per item by different users, naturally capturing diverse valid assembly orders. We carefully consider all potential assembly steps by manually inspecting the installation videos. In preparation, we labeled the sets of all possible correction options as answers. In the training stage, models are encouraged to generate all potential options, and, during evaluation, a prediction is marked as correct if it is a subset of the ground truth answer set.

- **Dynamic Part Tagging** Letter part tags ['A'-'Z'] are randomly assigned for the parts in each frame, i.e., the same part will have a different letter label in various questions, preventing the model from memorizing static associations between tags and parts, forcing it to focus on 3D structural features in the assembly context.

- **Shuffled Options** Option order is randomized to ensure the model relies on reasoning rather than positional bias

### 3.3 ASSEMBLY-V1: A BASELINE MODEL TRAINING WITH SUPERVISED FINE-TUNING

Treating Assembly-V1 as a baseline, we fine-tuned the Qwen2-VL-2B-Instruct Wang et al. (2024) model using Supervised Fine-Tuning (SFT). The fine-tuning was performed with the help of LlamaFactory Zheng et al. (2024). In the training procedure, we use the collected question/vision-answer pairs to form the chat template, and we apply the SFT function provided by the LlamaFactory to finish the training. More training parameters and details are discussed in the Appendix.

### 3.4 ASSEMBLY-R1: VISUAL REASONING USING REINFORCEMENT LEARNING

As stated before, our proposed FurniBench task is challenging. This task requires the model not only to recognize object categories, but also to understand deeper information, such as geometric structures and the 3D relationships among objects in the image.

To achieve this goal, we apply the powerful RL tool, Group Relative Policy Optimization (GRPO) Guo et al. (2025); Shao et al. (2024), to train our model. The objective function of GRPO can be described as follows:

$$\mathcal{J}_{GRPO}(\theta) = \mathbb{E}[q_v \sim P(Q_V), \{o_i\}_{i=1}^{G} \sim \pi_{\theta_{old}}(O|q_v)]$$

$$\frac{1}{G} \sum_{i=1}^{G} \left( \min \left( \frac{\pi_\theta(o_i|q_v)}{\pi_{\theta_{old}}(o_i|q_v)} A_i, \text{clip} \left( \frac{\pi_\theta(o_i|q_v)}{\pi_{\theta_{old}}(o_i|q_v)}, 1-\varepsilon, 1+\varepsilon \right) A_i \right) - \beta \mathbb{D}_{KL}(\pi_\theta||\pi_{ref}) \right), \quad (1)$$

$$\mathbb{D}_{KL}(\pi_\theta||\pi_{ref}) = \frac{\pi_{ref}(o_i|q_v)}{\pi_\theta(o_i|q_v)} - \log \frac{\pi_{ref}(o_i|q_v)}{\pi_\theta(o_i|q_v)} - 1, \quad (2)$$

where $q_v$ represents the sampled question and image set; $\{o_1, o_2, \cdots, o_i\}$ are the outputs sampled from the policy model $\pi_\theta$ or the old policy model $\pi_{old}$; $\varepsilon$ and $\beta$ are hyper-parameters; $A_i$ calculated from the rewards $\{r_1, r_2, \cdots, r_G\}$ through the following formula:

$$A_i = \frac{r_i - \text{mean}(\{r_1, r_2, \cdots, r_G\})}{\text{std}(\{r_1, r_2, \cdots, r_G\})}. \quad (3)$$

### 3.4.1 REWARD DESIGN

The reward design is the key to the success of GRPO training. Our goal is to provide a straightforward and effective reward signal that motivates the model's reasoning chain while solving challenging problems with precise answers.

**Format Reward**  The model must produce a reasoning chain followed by a final answer using the required tags:

$$r_{\text{fmt}} = \mathbb{1}\big[ o \equiv \texttt{<think>}\ o_{\text{reason}}\ \texttt{</think>}\ \texttt{<answer>}\ o_{\text{ans}}\ \texttt{</answer>} \big]$$

**Accuracy Reward**  If the predicted answer matches the reference, the model receives +1:

$$r_{\text{acc}} = \mathbb{1}\big[ \text{canon}(o_{\text{ans}}) = o_{\text{ref}} \big]$$

Here, $\text{canon}(\cdot)$ denotes canonicalization of the answer (e.g., normalization of case/whitespace and parsing to a valid option or set for multi-select).

**Pure-Coverage Reward (PCR)**  The reward above doesn't favor those partially correct answers over wrong answers. As a result, the 'spark' is not captured. To solve this, we provide graded credit for strictly correct subsets (and zero for any wrong options) for multi-select MCQ, defined as:

$$r_{\text{pcr}} = \begin{cases} \dfrac{|O_{\text{pred}}|}{|O_{\text{gt}}|}, & \text{if } O_{\text{pred}} \subseteq O_{\text{gt}} \text{ and } O_{\text{pred}} \neq \varnothing, \\ 0, & \text{otherwise.} \end{cases}$$

where $O_{\text{gt}}$ is the ground-truth option set; $O_{\text{pred}}$ is the deduplicated model's answer set .

With the objectives and rewards discussed above, we train a reasoning model that can provide both a reasoning procedure and a correct answer. We discuss the training parameters in our Appendix.

## 4 EXPERIMENTS

### 4.1 DATASET AND EVALUATION

We randomly select 15,000 QA pairs from the FurniQA training set to fine-tune both Assembly-V1 and Assembly-R1. For general evaluation, we use 1,500 QA pairs from FurniQA testing branch. To enable deeper analysis of the models' intrinsic abilities, we additionally classify tasks into two groups: those solvable through pure recognition and those that require spatiotemporal reasoning.

Furthermore, to assess the generalizability, we further evaluate both models on multiple OOD benchmarks, including GRiD-3D Lee et al. (2022), GQA Hudson & Manning (2019), CV-Bench Tong et al. (2024), SQA3D Ma et al. (2023), Super-CLEVR Li et al. (2023b). The performance of the candidate models is measured based on the accuracy of the answer responses.

## 4.2 HARDWARE AND IMPLEMENTATION DETAILS

In our experiments, we use $2 \times$ NVIDIA A100 80GB GPUs to train the models, including Assembly-V1 and Assembly-R1. For both training procedures, we set the per-device batch size to 1 and the gradient accumulation steps to 4. The training step is set to 1800. We tune all the parameters of the models at both the SFT and GRPO stages. Due to the page limit, we demonstrate more training details in our Appendix.

Table 1: Accuracy comparison across models and task categories on FurniBench (%).

| Models | PR[†] | PC[‡] | GAU[*] | Total |
|---|---|---|---|---|
| Gemini-2.5-Pro | 48.8 | 49.3 | 61.4 | 55.8 |
| GPT-4o | 33.4 | 24.0 | 56.6 | 45.7 |
| Qwen2-VL-7B-Instruct | 62.5 | 9.3 | 40.0 | 47.5 |
| LLaVA-1.5-7B-HF | 57.0 | 16.0 | 39.7 | 45.4 |
| Qwen2.5-VL-3B-Instruct | 41.6 | 13.3 | 38.2 | 38.3 |
| InternVL3-2B-Instruct | 32.4 | 16.0 | 35.4 | 33.3 |
| Qwen2-VL-2B-Instruct | 37.8 | 28.0 | 41.0 | 39.1 |
| Assembly-V1 | 68.9 | 44.0 | 72.4 | 69.6 |
| Assembly-R1 | **71.9** | **74.7** | **74.7** | **73.6** |

[†] PR: Part Recognition.    [‡] PC: Part Connectivity.    [*] GAU: General Assembly Understanding.
All values are accuracy in %.

## 4.3 QUANTITATIVE RESULTS

Table 1 reports accuracy across three task categories, Part Recognition (PR), Part Connectivity (PC), and General Assembly Understanding (GAU), alongside the overall performance Total. Table 2 summarizes accuracy for two capability-oriented groupings: Semantic Understanding, emphasizing recognition and global assembly semantics, and Spatial Reasoning, which are least solvable by pure 2D clues, requiring models' spatial-temporal understandings. The comparisons include seven open-source and closed-source commercial VLMs of similar or larger scales: GPT-4o OpenAI (2024a), Gemini-2.5-Pro Google DeepMind (2025), Qwen2-VL-7B-Instruct Wang et al. (2024),

Table 2: Benchmark Analysis: Semantic Understanding vs. Spatial Reasoning (%).

| Models | Semantic Understanding | Spatial Reasoning |
|---|---|---|
| Gemini-2.5-Pro | 59.2 | 47.6 |
| GPT-4o | 47.2 | 32.3 |
| Qwen2-VL-7B-Instruct | 44.9 | 20.1 |
| LLaVA-1.5-7B-HF | 41.1 | 21.9 |
| Qwen2.5-VL-3B-Instruct | 38.9 | 22.9 |
| InternVL3-2B-Instruct | 28.8 | 20.1 |
| Qwen2-VL-2B-Instruct | 34.5 | 27.1 |
| Assembly-V1 | 69.2 | 49.0 |
| Assembly-R1 | **72.2** | **59.7** |

All values are accuracy in %.

LLaVA-1.5-7B-Instruct Liu et al. (2023), Qwen2.5-VL-3B-Instruct Bai et al. (2025), InternVL3-2B-Instruct Zhu et al. (2025), and Qwen2-VL-2B-Instruct Wang et al. (2024), and our two in-domain models: Assembly-V1 and Assembly-R1.

### 4.3.1 Benchmarking for VLM baselines

As shown in Table 1, models with larger scale lead overall: Gemini-2.5-Pro attain a total accuracy of 55.8%, ahead of GPT-4o at 45.7%, while the strongest open-source 7B models, Qwen2-VL-7B-Instruct and LLaVA-1.5-7B-HF, reach 47.5% and 45.4%, respectively, outperforming other candidates with 2B/3B scale.

Category-wise analysis reveals distinct performance profiles. In Part Recognition, larger open-source models perform well, e.g., Qwen2-VL-7B-Instruct at 62.5%, indicating recognition tasks are tractable. For General Assembly Understanding, commercial models lead: Gemini-2.5-Pro at 61.4% and GPT-4o at 56.6%, reflecting superior semantic planning. However, Part Connectivity exposes a critical gap: requiring genuine geometric interpretation, most models fail significantly. While Gemini-2.5-Pro achieves 49.3%, others, like Qwen2-VL-7B-Instruct at 9.3%, struggle, highlighting that off-the-shelf VLMs lack the relational reasoning needed to transcend 2D perception.

A similar pattern is shown in Table 2. All models, especially for smaller-scale open-sourced VLMs, exhibit a substantial drop from Semantic Understanding to Spatial Reasoning, indicating that tasks that are least solvable by pure 2D cues remain challenging. For example, Qwen2-VL-7B-Instruct falls from 44.9% to 20.1%, LLaVA-1.5-7B-HF from 41.1% to 21.9%. Even the strongest commercial model achieves only 47.6% and 32.3% accuracy in tasks requiring complex spatial reasoning. These trends confirm that spatial–temporal reasoning, like interpreting geometric structure, relational constraints, and physical plausibility, is the principal bottleneck for general-purpose VLMs on FurniBench.

### 4.3.2 Comparisons with baselines

Although our in-domain models decisively outperform baselines across Tables 1 and 2, with the largest gains appearing in spatially demanding settings, important contrasts emerge between SFT and RL-based models. Overall, Assembly-R1 attains the best Total accuracy at 73.6%, surpassing Assembly-V1 at 69.6% and the strongest baseline, Gemini-2.5-Pro at 55.8%.

Category-wise, Assembly-R1 improves upon Assembly-V1 in all three categories, and delivers substantial margins over Gemini-2.5-Pro (PR: +23.1, PC: +25.4, GAU: +13.3). Although the SFT model Assembly-V1 improves overall over Gemini, its PC performance (44.0%) is weaker than Gemini's (49.3%), indicating that SFT primarily memorizes dataset-specific recognition and semantic patterns rather than learning intrinsic spatial reasoning.

By contrast, the RL-based Assembly-R1 closes the spatial gap: it not only lifts PC to 74.7% but also delivers strong gains on the capability-oriented spatial groups, with 10.7% gain for Assembly-V1 and 12.1% lead for Gemini-2.5-Pro, while maintaining improvements in semantic understanding. Taken together, these results suggest that reinforcement learning encourages the acquisition of geometric and relational constraints and yields superior generalization on tasks least solvable by pure 2D cues, whereas supervised fine-tuning alone is insufficient to overcome the spatial reasoning bottleneck.

### 4.4 Qualitative Results

We also present some qualitative results in Fig. 3 to showcase our trained models. Overall, the figure shows that Assembly-R1 achieves the best performance, while both the base model and Assembly-V1 struggle to provide the correct answers. Assembly-R1 can not only give the correct answers, but also outputs the detailed reasoning processes. Taking the Fig. 3 (a) as an example, Assembly-R1 can output both the reasons for selecting the correct answer and analysis for the incorrect answer. This indicates the deep analysis ability of the Assembly-R1. The SFT model Assembly-V1 cannot provide the analysis as the SFT tends to "remember" the correct answers. This phenomenon also demonstrates the superiority of the RL-based model. We can summarize that the RL training algorithm GRPO indeed can improve both the reasoning ability and the accuracy of the answer.

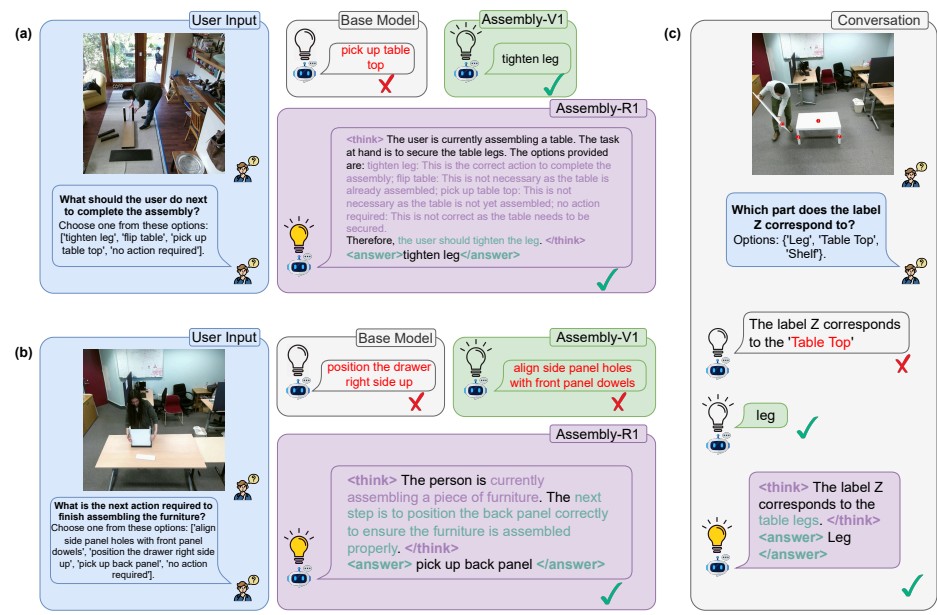

Figure 3: Qualitative results: Three example cases illustrated in (a), (b), and (c). Each example includes User Input (image + text, highlighted in blue), Base Model response (light gray), Assembly-V1 response (green), and Assembly-R1 response (purple). A tick or cross next to each model's response indicates correctness.

Table 3: Consecutive Assembly Step Planning Success Rate (%).

| Models | step@1 | step@2 | step@3 | step@4 | step@5 |
|---|---|---|---|---|---|
| Gemini-2.5-Pro | 49.0 | 25.5 | 13.1 | 6.9 | 2.9 |
| GPT-4o | 45.7 | 23.8 | 10.5 | 4.9 | 1.0 |
| Base Model | 26.3 | 4.4 | 0.7 | 0.0 | 0.0 |
| Assembly-V1 | 51.0 | 26.5 | 12.8 | 7.8 | 2.9 |
| Assembly-R1 | **58.4** | **36.5** | **23.9** | **15.2** | **9.8** |

All values are success rate in %. Success at step@$k$ requires correctly planning the next $k$ consecutive assembly steps.

## 4.5 APPLICATIONS TO EMBODIED AI

To assess adaptability to Embodied AI, we construct a subset task, namely Consecutive Assembly Step Planning, where success at depth $k$ requires correctly planning the next $k$ consecutive assembly steps. As shown in Table 3, Assembly-R1 achieves the highest success rate across all depths, while Assembly-V1 matches or slightly exceeds commercial baselines at shallow depths but decays more rapidly with increasing step count. In contrast, general-purpose VLMs fail in multi-step assembly planning: Gemini-2.5-Pro falls from 49.0% to 2.9% by step@5, and GPT-4o from 45.7% to 1.0%, reflecting compounding planning errors when reasoning over multi-step geometric dependencies.

These results confirm that our method scales reliably to multi-step spatial-temporal reasoning, preserving plan consistency over longer horizons where baselines falter. Assembly-R1 thus establishes a viable pathway for high-level robotic planning under spatial constraints.

## 4.6 ABLATIONS

We conduct ablation studies on reward design and training strategies to validate our design, as shown in Table 4. The first design is Assembly-V1 + RL post-training, and the second only uses format reward and accuracy rewards. Results show our design has the best overall performance.

Table 4: Ablation: RL reward variants on FurniBench (%).

| Models | PR[†] | PC[‡] | GAU[*] | Total |
|---|---|---|---|---|
| Assembly-V1 w/ RL | 69.1 | 23.1 | 70.7 | 67.6 |
| Assembly-R1 w/o PCR | **73.4** | 32.0 | 73.8 | 71.5 |
| Assembly-R1 w/ PCR (ours) | 71.9 | **74.7** | **74.7** | **73.6** |

[†] PR: Part Recognition.  [‡] PC: Part Connectivity.  [*] GAU: General Assembly Understanding.
All values are accuracy in %.

### 4.7 OUT-OF-DOMAIN RESULTS

We present zero-shot Out-Of-Domain (OOD) evaluations on GRiD-3D Lee et al. (2022), GQA Hudson & Manning (2019), CV-Bench Tong et al. (2024), SQA3D Ma et al. (2023), and Super-CLEVR Li et al. (2023b), as shown in Table 5. Assembly-R1 demonstrates consistent gains over the Base Model across all benchmarks (e.g., GRiD-3D +6.3%, GQA +4.5%), whereas the SFT model Assembly-V1 underperforms the Base Model on all five datasets, highlighting the catastrophic forgetting of pure SFT under distribution shift. In contrast, Assembly-R1 achieves substantial margins over SFT on spatially demanding tasks like GRiD-3D (+7.2%) and SQA3D (+13.5%), indicating that reinforcement learning fosters robust spatial–temporal inference rather than surface-level pattern matching. Notably, despite being trained primarily on multiple-choice questions (MCQ), Assembly-R1 generalizes effectively to open-vocabulary (free-text) tasks such as GRiD-3D and SQA3D. These findings support the "SFT Memorizes, RL Generalizes" hypothesis Chu et al. (2025): while SFT aligns models to specific training distributions, RL equips them with transferable decision rules for broader applicability.

Table 5: Evaluation on OOD Benchmarks, including Embodied AI-specific Benchmarks (%).

| Models | GRiD-3D | GQA | CV-Bench | SQA3D | Super-CLEVR |
|---|---|---|---|---|---|
| Base Model | 34.4 | 61.9 | 62.4 | 32.2 | 42.7 |
| Assembly-V1 | 33.5 | 55.3 | 49.6 | 21.8 | 41.2 |
| Assembly-R1 | **40.7** | **66.4** | **64.5** | **35.3** | **45.9** |

Task formats: GRiD-3D (Free Text), GQA (Free Text), CV-Bench (MCQ), SQA3D (Free Text), Super-CLEVR (Free Text). All values are accuracy in %.

### 4.8 LIMITATIONS

Despite the richness and large scale of our FurniQA, there is still room for improvement in terms of diversity. Specifically, the dataset could benefit from incorporating a broader range of QA task types and assembly objects, more diverse camera shooting angles, input modalities, like depth information, and indoor/outdoor assembly scenes. Enhancing the dataset in these aspects could assist the model to generalize better to real-world applications and unseen configurations.

## 5 CONCLUSION

In conclusion, we propose a new benchmark, FurniBench, along with a new dataset, FurniQA, to assess the 3D structural and spatial understanding of large models. We also trained new large models, Assembly-V1 and Assembly-R1, based on our dataset. We successfully established our new benchmark by testing our trained models and other open-source VLMs. In addition, we use out-of-domain experiments to demonstrate the phenomenon of "SFT Memorizes, RL Generalizes." In the future, we plan to test our models in real industrial environments, such as industrial robotic assembly scenarios, Embodied AI, etc.

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

# A    APPENDIX

## A.1    SUPERVISED FINE-TUNING (SFT) CONFIGURATIONS

The SFT training configurations are listed in Table 6. The fine-tuning was performed with the help of LlamaFactory Zheng et al. (2024). The dataset is structured in Alpaca format  Taori et al. (2023) for training the model.

| Parameter | Value |
| --- | --- |
| model_name_or_path | `Qwen/Qwen2-VL-2B-Instruct` |
| trust_remote_code | `true` |
| stage | `sft` |
| do_train | `true` |
| finetuning_type | `full` |
| freeze_vision_tower | `false` |
| freeze_multi_modal_projector | `false` |
| freeze_language_model | `false` |
| deepspeed | `LLaMA-Factory/examples/deepspeed/ds_z3_config.json` |
| dataset | `FurniBench_train_shuffled_selected_15000` |
| template | `qwen2_vl` |
| cutoff_len | `20480` |
| preprocessing_num_workers | `16` |
| dataloader_num_workers | `4` |
| output_dir | `outputs/qwen2_vl-2b_512_15000/sft` |
| logging_steps | `25` |
| save_steps | `300` |
| report_to | `wandb` |
| batch_size | `8` |
| learning_rate | `5.0e-5` |
| num_train_epochs | `3` |
| lr_scheduler_type | `cosine` |
| warmup_ratio | `0.1` |
| bf16 | `true` |
| ddp_timeout | `180000000` |
| resume_from_checkpoint | `null` |

Table 6: Supervised Fine-Tuning (SFT) & DeepSpeed training configurations.

## A.2 GROUP RELATIVE POLICY OPTIMIZATION (GRPO) CONFIGURATIONS

The GRPO model fine-tuning configurations are listed in Table 7. The multi-GPU training benefits from DeepSpeed Rasley et al. (2020).

| Parameter | Value |
|---|---|
| config_file | configs/zero2.yaml |
| model_name_or_path | Qwen/Qwen2-VL-2B-Instruct |
| dataset_name | FurniBench_train_shuffled_selected_15000 |
| max_prompt_length | 1024 |
| max_completion_length | 700 |
| learning_rate | 1.0e-6 |
| batch_size | 8 |
| logging_steps | 1 |
| bf16 | true |
| gradient_checkpointing | true |
| num_train_epochs | 3 |
| save_steps | 300 |
| save_only_model | true |
| report_to | wandb |
| compute_environment | LOCAL_MACHINE |
| distributed_type | DEEPSPEED |
| deepspeed_multinode_launcher | standard |
| zero_stage | 2 |
| zero3_init_flag | false |
| offload_optimizer_device | none |
| offload_param_device | none |
| mixed_precision | bf16 |
| downcast_bf16 | no |
| num_processes | 8 |
| num_machines | 1 |
| machine_rank | 0 |
| main_training_function | main |
| main_process_port | 44326 |
| rdzv_backend | static |
| same_network | true |
| use_cpu | false |
| tpu_use_cluster | false |
| tpu_use_sudo | false |
| tpu_env | [ ] |

Table 7: GRPO & DeepSpeed training configuration.

## A.3 DATASET - QUESTION REPRESENTATIVE EXPRESSIONS

FurniQA includes 15 distinct QA task types. To make the dataset more diverse, each task is associated with three representative question expressions, as illustrated in Table 8. When generating QA pairs for each assembly video frame, one of the three expressions for the corresponding question type is randomly selected.

| Question Type | Representative Expressions |
|---|---|
| **Single Part Recognition (MCQ)** | What is the part labeled in `{id}`? 
 Please identify the part labeled as `{id}`. 
 Which part does the label `{id}` correspond to? |
| **Part Set Completeness (YN)** | Are the currently labeled parts sufficient to complete the assembly? 
 Do the labeled parts cover everything needed for assembly? 
 Are all necessary parts labeled for assembly? |
| **Missing Part Recognition (MCQ)** | What other parts are required to complete the assembly? 
 Are there any parts not labeled that are needed? 
 Which parts are still required to finish the assembly? |
| **First Assemble Pair (MCQ)** | Which two parts can be assembled first? 
 Out of the listed pairs, which can be assembled at the beginning? 
 Select the pair of parts that should be assembled first. |
| **First Assemble Pair (YN)** | Can I directly attach Part A to Part B? 
 Are Part A and Part B ready to be connected now? 
 Is it possible to assemble them together now? |
| **Connection After Installation (MCQ)** | What parts does Part A connect to after installation? 
 After assembly, which parts will be connected to Part A? 
 Select the parts that will be attached to Part A. |
| **Disassemble First (MCQ)** | Which parts can be disassembled first? 
 Out of the listed parts, which can be removed first? 
 Select the part(s) that should be taken apart first. |
| **Object Recognition (MCQ)** | What could be the type of furniture? 
 What is the most likely furniture type? 
 Which furniture category do these parts belong to? |
| **Installation Completed (YN)** | Is the installation completed? 
 Has the assembly process finished? 
 Are all parts fully assembled now? |
| **Action Recognition (MCQ)** | What is the user doing in this frame? 
 Describe the action performed by the user. 
 Which activity is the user engaged in now? |
| **Action Recognition (YN)** | Is the user manipulating a `{part}`? 
 Is the user interacting with a `{part}`? 
 Do you see the user handling a `{part}`? |
| **Next Step Inference (MCQ)** | What should the user do next to complete the installation? 
 What is the next action required? 
 Which step should be performed next? |
| **Installation Preparation (MCQ)** | What should the user do next to prepare? 
 Which preparation is needed before continuing? 
 What action should be taken before the next step? |
| **Installation Assembly (MCQ)** | What should the user do next to complete the assembly? 
 Which assembly action comes next? 
 What is the next step in the assembly process? |
| **Ready for Installation (YN)** | Are the `{part}` ready to be installed? 
 Can the `{part}` be installed now? 
 Is any step required before installing the `{part}`? |

Table 8: Overview of all 15 question types in FurniQA with representative expressions. Each type has 3 variations to encourage language diversity. For easier evaluation, each question in the dataset comes with a list of options, either a list of different choices or Yes/No. To maintain clarity, the answer options are not shown in this table.

## A.4 DATASET - STATISTICS OF FURNIQA

Table 9: Statistics of FurniQA, including the main category, sub-category, task type, and quantities of corresponding QA pairs.

| Main Category | Sub Category | Type | Quantity |
|---|---|---|---|
| **Part Recognition** | Single Part Recognition | MCQ | 176,903 |
| | Part Set Completeness | YN | 176,903 |
| | Missing Part Recognition | MCQ | 154,105 |
| | Object Recognition | MCQ | 154,105 |
| **Part Connectivity** | First Assemble Pair | MCQ | 3,050 |
| | First Assemble Pair | YN | 10,654 |
| | Connection After Installation | MCQ | 45,786 |
| | First Dissemble Part | MCQ | 22,798 |
| **General Assembly Understanding** | Installation Completion | YN | 176,903 |
| | Action Recognition | MCQ | 150,286 |
| | Action Recognition | YN | 176,903 |
| | Next Step Inference | MCQ | 176,903 |
| | Installation Preparation | MCQ | 107,972 |
| | Installation Assembly | MCQ | 45,655 |
| | Ready For Installation | YN | 35,969 |

## A.5 Model Performance Comparisons

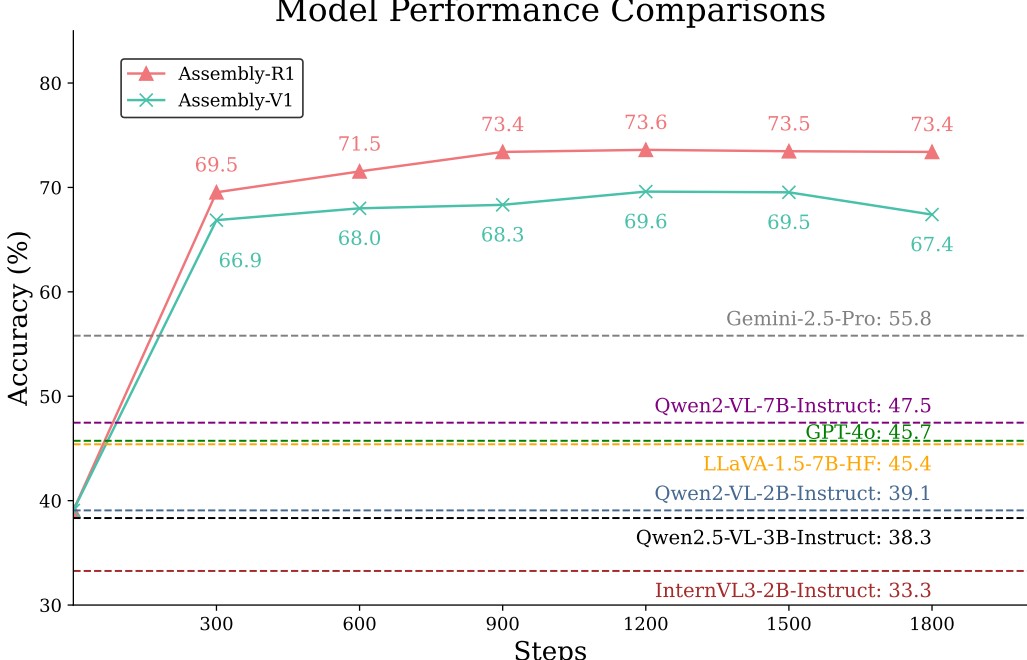

Figure 4: Performance comparison of various models on FurniBench. The green and red lines depict the progression of Assembly-V1 and Assembly-R1 performance throughout the training steps. Horizontal dashed lines indicate the benchmark performance of popular open-source vision-language models (VLMs).

## A.6 Additional Ablation Studies

**Reward Design Analysis.** To validate our reward structure, we explored alternative designs intended to explicitly encourage reasoning via chain length and logical structure. We tested two configurations: (1) a *Length-Incentivized Reward* ($r_{len} \propto N_{tokens}$), and (2) a *Logic-Keyword Reward* combining clipped length incentives with bonuses for transition words (e.g., "therefore", "next"). Both approaches led to severe reward hacking and training instability. The length-based reward caused a "verbosity explosion," where completion lengths surged to over 400 tokens as the model learned to filibuster rather than reason. Similarly, the keyword incentive encouraged repetitive, long-winded generation to maximize keyword frequency, significantly increasing compute costs without improving accuracy. These findings confirm that heuristic-based rewards induce superficial verbosity. Consequently, we retained our final design—a simple format constraint combined with a strong outcome-based accuracy reward—which allows the model to self-discover optimal reasoning patterns without bias.

## A.7 Additional Analysis - Reward Hacking

The experiment is conducted on the classic Assembly-R1 design, where there was no Pure Coverage Reward (PCR).

Reward hacking occurs when an agent exploits flaws in the reward design to gain rewards through unintended behaviors Shen et al. (2025). Zhou et al. (2025) show that rewarding reasoning length can lead models to generate longer outputs without improving reasoning quality.

Although we don't explicitly reward reasoning length, we still observe signs of reward hacking during training. We define the length reward hacking as a response that repeats with meaningless reasoning

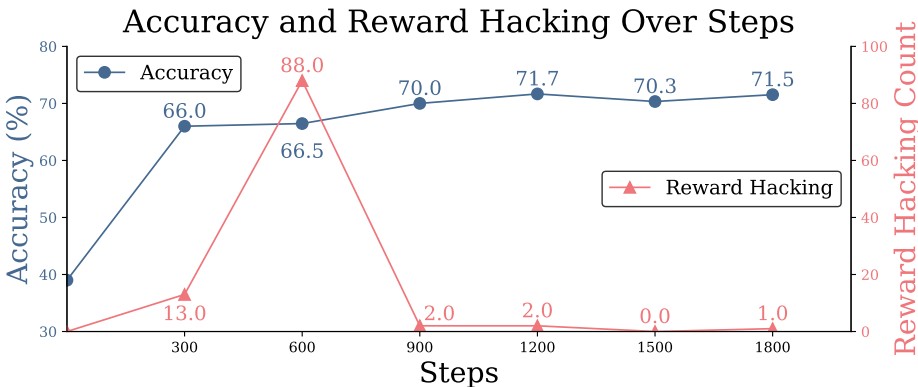

Figure 5: Visualization of model performance (blue line) and the number of reward hacking instances (red line) across training steps.

until reaching the output limit (1024) without a closing `</think>` tag. These incomplete responses suggest the model tries to exploit perceived reward signals without understanding the task.

As shown in Fig. 5, accuracy initially rises from 39.1% to 66.0% at 300 steps, with 13 reward hacking instances out of 1500 testing samples. This initial improvement in accuracy likely results from introducing the reasoning pattern, which the base model lacks. However, from 300 to 600 steps, hacking behavior increases while accuracy stagnates. In other words, the agent is optimizing for quantity over quality, generating longer but ineffective reasoning sequences. After 600 steps, rewards hacking diminishes and accuracy improves, reaching 71.7%. This is expected since our reward design does not explicitly favor long reasoning but rather meaningful thinking and accuracy. Eventually, the model shifts towards generating useful reasoning to gain more **Accuracy Reward**.

This observation highlights the importance of careful reward design in the RL-based fine-tuning framework for enhancing LLM/VLM reasoning capability.

## A.8 ADDITIONAL ANALYSIS - AVERAGE RESPONSE LENGTH

The experiment is conducted on the classic Assembly-R1 design, where there was no Pure Coverage Reward (PCR).

Fig. 6 shows the relationship between the model's performance and its average reasoning length over training steps. Importantly, we exclude samples flagged as reward hacking behavior when calculating the average reasoning length per response, so the statistics reflect only valid reasoning sequences.

Since our reward function does not explicitly encourage longer reasoning, the average length does not increase monotonically during training. Instead, it fluctuates between 17 and 35 words from step 300 onward. Notably, we can observe that the improvement in accuracy is usually accompanied by longer reasoning, while the periods of stable accuracy often show a decrease in reasoning response length.

In the early training phase, from the start to step 300, accuracy improves from 39.1% to 66.0%, with the average reasoning length reaching 39.1 tokens. Between steps 300 and 600, accuracy remains steady while the average reasoning length drops to 17.2 tokens. Then, from step 600 to step 1200, the accuracy climbs further to 71.1%, accompanied by an increment in average reasoning length to 35.1 tokens. Afterward, while the model keeps the accuracy around 71%, the average length decreases by over 10 tokens per response.

In summary, while the model is not directly rewarded for longer reasoning, it learns to use a more elaborate self-reflective reasoning chain to gain reward by improving answer accuracy. At the same time, it continues to refine its reasoning pattern to avoid unnecessary verbosity.

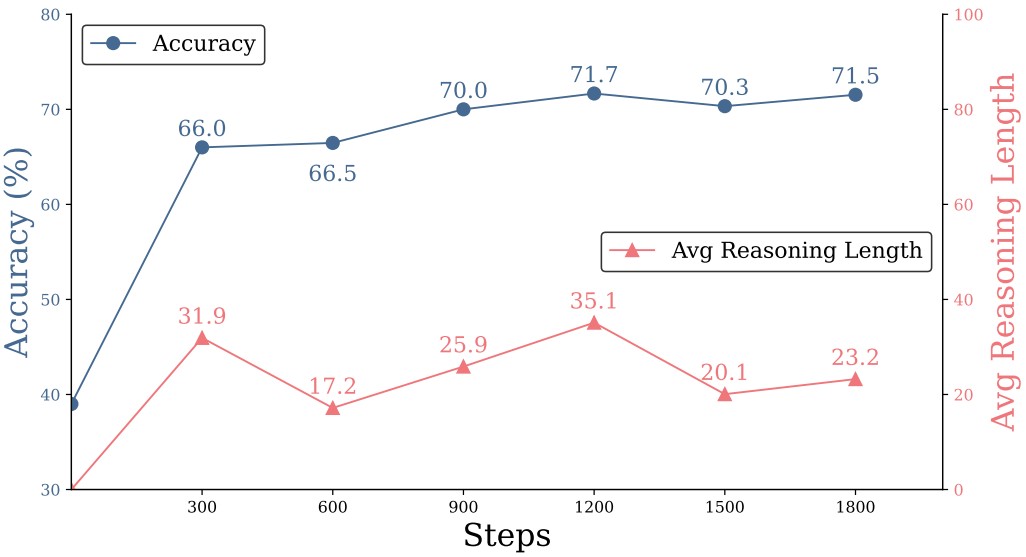

Figure 6: Visualization of model performance (blue line) and the average length of reasoning (red line) across training steps

