# OpenReview forum: "Assembly-R1: 3D Assembly Reasoning via RL-based Vision Language Models"
_ICLR.cc/2026/Conference — Submitted to ICLR 2026_

### Official Review · Reviewer_tmox · 2025-11-01

**Soundness:** 2
**Presentation:** 2
**Contribution:** 2
**Rating:** 6
**Confidence:** 3

**Summary:**

This paper introduces FurniBench, a benchmark for Visual Question Answering (VQA) in 3D assembly tasks, along with FurniQA, a dataset containing 1.6M QA pairs derived from the IKEA ASM dataset. The authors establish a supervised fine-tuning (SFT) baseline (Assembly-V1) and propose Assembly-R1, trained using Group Relative Policy Optimization (GRPO).

**Strengths:**

1. The motivation for creating a benchmark in 3D assembly reasoning is valid, as general VQA benchmarks often lack the required complexity in structural and procedural understanding.
2. The approaches to reduce bias and subjectivity in dataset construction are reasonable.
3. The GRPO-based Assembly-R1 shows significant improvements over both the base model (+32.6%) and SFT baseline (+7.6%).
4. The out-of-domain evaluation on CVBench (Table 2) demonstrates the generalization ability brought by RL.

**Weaknesses:**

1. While FurniBench claims to assess 3D structural and spatial reasoning, the tasks seem to primarily focus on 2D part recognition and sequence inference. Key 3D spatial challenges like reasoning about occlusion, perspective changes, or spatial relationships are not explicitly evaluated. Many tasks could potentially be solved using 2D visual cues and logic, without requiring 3D understanding.
2. The "Part Connectivity" category shows minimal improvement (28.0% → 32.0%), suggesting the model may still struggle with 3D structural relationships.
3. Only one base model architecture (Qwen2-VL-2B) is fine-tuned. It would be valuable to see results on other architectures to demonstrate the general applicability. Furthermore, the paper does not compare with other reasoning enhancement techniques, making it difficult to assess its relative advantages.

**Questions:**

1. Why is the depth information in the IKEA ASM dataset not utilized in dataset construction?
2. Could you provide more detailed error analysis, particularly for the "Part Connectivity" tasks where improvement is minimal?
3. Given Weakness 1, could you identify a subset of questions that are most indicative of 3D spatial reasoning (i.e., those least solvable by 2D cues alone)? Reporting the model's performance specifically on this challenging subset would provide a clearer measure of its true 3D capabilities
4. The dataset construction mentions that the questions are "manually calibrated." Could you provide more details on this process, such as the calibration procedure and the total annotation time involved?

---

> ### Author Response · Authors · 2025-11-21
>
> ## **Reply to Reviewer tmox**
>
> We appreciate the reviewer's thoughtful and constructive review! Below, we address the concerns with additional experimental results and analysis.
>
> ### **Part Connectivity: Minimal Gain and Error Analysis**
>
> **Response to Weakness 2, Question 2**
>
> We appreciate the reviewer's careful observation regarding the minimal gain on the Part Connectivity Task. We agree that Part Connectivity is the most challenging category because it demands genuine structural reasoning (topology, feasible attachments, and disassembly ordering).
>
> To address this, we provide a comprehensive comparison between our model and other baselines, specifically highlighting the improvements in Part Connectivity with our updated reward design.
>
> |Task Category|Part Recognition|Part Connectivity|General Assembly Understanding|Total|
> |--|--|--|--|--|
> |**External Baselines**|||
> |GPT-4o|33.4|24.0|56.6|45.7|
> |Gemini-2.5-pro|48.8|49.3|61.4|55.8|
> |Qwen2-VL-7B-Instruct|62.5|9.3|40.0|47.5|
> |LLaVA-1.5-7B-HF|57.0|16.0|39.7|45.4|
> |Qwen2.5-VL-3B-Instruct|41.6|13.3|38.2|38.3|
> |InternVL3-2B-Instruct|32.4|16.0|35.4|33.3|
> |Qwen2-VL-2B-Instruct|37.8|28.0|41.0|39.1|
> |**Our Models**|||
> |Assembly-V1 (Original) |67.7|28.0|64.3|63.9|
> |Assembly-V1 (Updated) |67.4|36.0|71.9|68.3|
> |Assembly-R1 (Original) |**73.4**|32.0|**73.8**|**71.5**|
> |Assembly-R1 (Updated) |70.4|**62.7**|70.9|70.3|
>
> **Table 1:** Comprehensive accuracy comparison of Assembly-R1 against open-source and closed-source baselines across different task categories on FurniBench.
>
> *Note: Original model uses the original reward design and config. Updated V1 uses a new learning rate, and Updated R1 uses a new accuracy reward design, which will be introduced below.*
>
> **Analysis:**
> 1.  **Baseline Comparison:** Most baselines struggle with Part Connectivity, with the notable exception of Gemini-2.5-Pro (49.3%).
> 2.  **Root Cause of Initial Low Performance:** We found that many Part Connectivity questions have multiple correct answers (e.g., multiple valid connections). For each QA pair, we constructed a ground-truth answer set to reduce bias from forcing a single answer. However, our original reward treated correctness as “all-or-nothing” during training, so partially correct responses received no credit—making them indistinguishable from incorrect ones and impeding incremental learning.
> 3.  **Solution (Graded Reward):** We revised the accuracy rewards to provide graded credit to partially correct, strictly subset answers, while still penalizing any wrong options.
>
>     Specifically, let $A_{gt}$ be the ground-truth set of correct options and $A_{pred}$ the model's chosen set. **If** $A_{pred} \subseteq A_{gt}$ (model's answer is a subset of ground truth): $$r_{acc} = \frac{|A_{pred}|}{|A_{gt}|}$$ **If** $A_{pred} \not\subseteq A_{gt}$ (contains any incorrect option) **or** $A_{pred} = \emptyset$ (empty answer): $$r_{acc} = 0$$
>
>     *For example, if the ground truth set is ['A', 'I', 'O', 'T', 'E'], model's response of ['O', 'A', 'E'] will be rewarded 0.6, and model's response of [] or ['A', 'B', 'O'] will be rewarded 0.*
>
>     This is reasonable because understanding part connectivity is the key to installing the parts correctly. Partially correct answers still push towards correct installation just with fewer assembly combinations, but wrong answers may result in structural misplacement. Meanwhile, we keep the format reward (+1) for producing valid response structures.
>
> 4.  **Outcome:** With this new accuracy reward design, the model is rewarded for partially correct responses, allowing it to learn connectivity in smaller, verifiable steps. Qualitatively, the model now avoids “all-or-nothing” behavior and exhibits more consistent structural reasoning. After applying this new reward, the performance of Assembly-R1 on the Part Connectivity Task improved significantly to **62.7%**, outperforming all other baselines, including Gemini-2.5-Pro. This indicates that our model has improved capability in structural understanding.

---

> ### Author Response · Authors · 2025-11-21
>
> ### **Clarification of "3D" Reasoning and Challenging Subset**
>
> **Response to Weakness 1, Question 3**
>
> By "3D", we mean structural and spatial reasoning required for furniture assembly, which requires the model to understand contact relationships, readiness for connections, and step feasibility, rather than pure 2D object recognition. More specifically:
>
> 1.  **Missing Part Recognition:** Explicitly tests occlusion and partial labeling scenarios. The model must infer whether any critical parts are missing by reasoning over the visible labeled set and assembly constraints.
> 2.  **Connection After Installation:** The model is expected to infer the structure of the furniture given a part set where they are not/partially installed. The model needs to reason about structural constraints before the complete shape is formed.
> 3.  **First Disassemble Part:** Requires the model to reason about which components are removable given the current state and physical constraints.
> 4.  **Next Step Inference:** Requires sequencing conditioned on physical feasibility (e.g., alignment readiness), going beyond 2D features.
>
> **Performance on Challenging Subset:** As shown in **Table 1**, our updated Assembly-R1 achieves **62.7%** on Part Connectivity and **70.9%** on General Assembly Understanding, demonstrating robust 3D capabilities. Furthermore, our OOD results on **GRiD-3D** [1] (a purely 3D spatial task) show a significant gain (**42.7%** vs 34.4% for Base), confirming the transfer of 3D reasoning skills.
>
> #### **Challenging 3D Subset**
>
> **Response to Question 3**
>
> We agree with the reviewer's idea of developing a subset of questions that are most indicative of 3D spatial reasoning.
>
> We identify the following tasks as the "Challenging 3D Subset" given the reasons mentioned in the section above:
>
> * Missing Part Recognition
> * Connection After Installation
> * First Assemble
> * First Disassemble
> * Next Step Planning
> * Installation
>
> The table below shows the performance of our models and baseline.
>
> |Model|3D subset|
> |--|--|
> |GPT-4o|30.8|
> |Gemini-2.5-Pro|45.2|
> |Qwen2-VL-7B-Instruct|40.7|
> |LLaVA-1.5-7B-HF|29.0|
> |Qwen2.5-VL-3B-Instruct|42.9|
> |InternVL3-2B-Instruct|28.3|
> |Qwen2-VL-2B-Instruct|35.0|
> |**Our Models**|
> |Assembly-V1 (Original) |57.6|
> |Assembly-V1 (Updated) |62.0|
> |Assembly-R1 (Original) |64.3|
> |Assembly-R1 (Updated) |**69.5**|
>
> **Analysis:**
> 1.  **Increased Difficulty:** As shown in the table, the performance of all baseline models on this "Challenging 3D Subset" is consistently lower than their performance on the complete dataset (e.g., GPT-4o drops from 45.7% to 30.8%, and Qwen2-VL-2B drops from 39.1% to 35.0%). This confirms that this subset effectively isolates tasks requiring deeper spatial reasoning, which are less solvable by simple 2D cues.
> 2.  **Scaling Law Validation:** We observe that newer and larger models generally perform better on this subset (e.g., Qwen2.5-VL-3B > Qwen2-VL-2B, and Gemini-2.5-Pro performs the best, achieving 45.2%). This trend suggests that the tasks in this subset are well-posed and solvable with increased model capacity and reasoning capability.
> 3.  **Superiority of Assembly-R1:** Our Assembly-R1 (Updated) achieves the highest accuracy of **69.5%**, significantly outperforming all baselines and our SFT version. Notably, the improvement from the updated reward design is evident here, demonstrating that our method effectively teaches the model to handle complex and challenging 3D spatial relationships.
>
> ### **Dataset Calibration**
>
> **Response to Question 4**
>
> To address the question regarding the manual calibration process, we employed a rigorous multi-stage validation process:
>
> 1.  **Source Validity:** Our dataset is derived from IKEA-ASM, which contains real-world assembly videos. We manually verified that the assembly sequences in our ground truth align with physical feasibility.
> 2.  **Human-in-the-Loop Calibration:** For ambiguous tasks like "Part Connectivity" (where multiple valid connections exist), we conducted manual inspections to construct robust ground-truth sets that include all valid permutations. This prevents the "false negative" issue where a valid but alternative assembly step is penalized.
> 3.  **Question Phrasing Verification:** We manually reviewed a subset of generated questions to ensure they are unambiguous and contextually appropriate (e.g., ensuring "First Disassemble Part" questions are only asked when the furniture is in a state that allows disassembly).

---

> ### Author Response · Authors · 2025-11-21
>
> ### **Design Selection**
>
> **Response to Weakness 3**
>
> We chose Qwen2-VL-2B-Instruct for fine-tuning due to resource constraints and its strong performance/efficiency balance. This makes it a suitable candidate for comparing different strategies (SFT vs. GRPO) and benchmarking against other open-source and closed-source commercial VLMs.
>
> ### **Ablation Study - Reward Design**
>
> **Response to Weakness 3**
>
> To address the concern regarding reward sensitivity and design choices, we conducted ablation studies with alternative reward structures to encourage reasoning. Specifically, we explored explicit incentives for chain length and logical structure. However, we found that these heuristic-based rewards led to instability and reward hacking, reinforcing our decision to rely on our reward design.
>
> We tested two alternative configurations:
>
> 1.  **Length-Incentivized Reward:**
>     *   **Design:** Added a term $r_{len} = N_{tokens} \cdot 0.001$ (capped at 700 tokens $\approx$ 0.7 reward) to encourage longer chains.
>     *   **Outcome:** This caused a "verbosity explosion." The average completion length surged to 200 tokens by step 40 and exceeded 400 tokens by step 50. The model learned to "filibuster" to maximize $r_{len}$ rather than improve reasoning, leading to training instability and significantly increased compute costs without accuracy gains.
>
> 2.  **Logic-Keyword & Clipped-Length Reward:**
>     *   **Design:** Added a clipped length reward ($min(N_{tokens} \cdot 0.005, 0.25)$) and a "Logic Reward" triggered by keywords (e.g., *therefore, thus, first, next*).
>     *   **Outcome:** Although the length reward was capped, the keyword incentive indirectly encouraged verbosity. The model generated repetitive, long-winded chains to maximize the probability of hitting logic keywords. Completion length reached 400 tokens by step 70, again resulting in instability and slow convergence.
>
> **Conclusion:** These ablations demonstrate that **explicitly rewarding length or specific tokens induces reward hacking** (verbosity and repetition) rather than genuine reasoning. The increase in completion length also results in excessive training time and resource usage. This validates our final design: using a simple format constraint combined with a strong, outcome-based accuracy reward allows the model to self-discover optimal reasoning patterns without heuristic bias.
>
> ### **Not Using Depth**
>
> **Response to Question 1**
>
> Thanks for pointing this out; we did not include depth because this work focuses on assembly-oriented spatial reasoning from RGB, which keeps VQA evaluation fair and comparable and isolates our training/reward effects, while preserving transfer to RGB-only OOD benchmarks; we will consider to add depth in future work.
>
> ## References
>
> [1] Lee, J. H., Kerzel, M., Ahrens, K., Weber, C., & Wermter, S. (2022). *What is Right for Me is Not Yet Right for You: A Dataset for Grounding Relative Directions via Multi-Task Learning*. In Proceedings of the International Joint Conference on Artificial Intelligence (IJCAI).

---

### Official Review · Reviewer_85gg · 2025-11-01

**Soundness:** 3
**Presentation:** 2
**Contribution:** 3
**Rating:** 4
**Confidence:** 3

**Summary:**

1. This paper introduces FurniBench, a benchmark for 3D part-assembly visual question answering (VQA), and its associated dataset FurniQA, which contains 1.6 M QA pairs generated from the IKEA-ASM dataset.

2. It also proposes Assembly-R1, a reasoning-enhanced Vision-Language Model (VLM) trained using Group Relative Policy Optimization (GRPO), inspired by DeepSeek-R1, to improve structural and spatial reasoning.

3. Compared with the base model (Qwen2-VL-2B-Instruct, 39.1 %) and an SFT baseline (Assembly-V1, 64.9 %), the GRPO-based model reaches 71.7 % accuracy on FurniBench and 63.5 % on CVBench, showing that reinforcement learning can strengthen reasoning generalization.

**Strengths:**

1. Novel benchmark contribution: FurniBench fills an under-explored gap in 3D part-assembly reasoning for VQA, offering a structured taxonomy (part recognition, connectivity, and assembly reasoning).

2. Methodological clarity: The paper provides clear formulation of supervised (SFT) and RL (GRPO) stages and the trainig pipelines, and quantitative ablations demonstrate measurable gains, showing how the .

3. Empirical completeness: Experiments include both in-domain and out-of-domain evaluations (on CV-BENCH), along with reward-hacking analysis, adding credibility to the claims that “SFT memorizes, RL generalizes.”

**Weaknesses:**

1. While the results of baseline models (e.g. Gemini 2.5 Pro, GPT-4o,) are shown in the Figure 3, it would be clearer and more informative to include the numerical baseline performance in Table 1 for easy reference. This would help readers quickly grasp the comparative results on the new benchmark. In addition, the accuracy curve plot (Figure 3) currently provides limited insight and could be moved to the appendix to make space for more comprehensive quantitative tables in the main text.

2. Lack of human evaluation or qualitative validation: No human study or inter-annotator validation is provided to verify the realism or difficulty of the generated QA pairs.

**Questions:**

1. Benchmark Clarity and Statistics. The benchmark design section is quite detailed, but it would be helpful to include statistical visualizations (e.g., distributions of task types, question difficulty levels, or part categories). Could the authors provide these quantitative breakdowns to better illustrate the dataset’s novelty and coverage?

2. How do the authors believe the CoT process contributes to part recognition rather than simply restating the answer?
For example, in Figure 4(c), the reasoning step “\<think\> The label Z corresponds to the table legs. \</think\>” appears to directly repeat the answer rather than providing additional reasoning.

---

> ### Author Response · Authors · 2025-11-21
>
> ## **Reply to Reviewer 85gg**
>
> We appreciate the reviewer's thoughtful and constructive review! Below, we address the concerns with additional experimental results and analysis.
>
> ### **Comprehensive Baseline Comparison**
>
> **Response to Weakness 1**
>
> We agree with the reviewer that a numerical table is more informative than the accuracy curve. We will move the accuracy curve (Figure 3) to the appendix and include the following comprehensive comparison table in the main text.
>
> |Task Category|Part Recognition|Part Connectivity|General Assembly Understanding|Total|
> |--|--|--|--|--|
> |**External Baselines**||||
> |GPT-4o|33.4|24.0|56.6|45.7|
> |Gemini-2.5-Pro|48.8|49.3|61.4|55.8|
> |Qwen2-VL-7B-Instruct|62.5|9.3|40.0|47.5|
> |LLaVA-1.5-7B-HF|57.0|16.0|39.7|45.4|
> |Qwen2.5-VL-3B-Instruct|41.6|13.3|38.2|38.3|
> |InternVL3-2B-Instruct|32.4|16.0|35.4|33.3|
> |Qwen2-VL-2B-Instruct|37.8|28.0|41.0|39.1|
> |**Our Models**||||
> |Assembly-V1 (Original) |67.7|28.0|64.3|63.9|
> |Assembly-V1 (Updated) |67.4|36.0|71.9|68.3|
> |Assembly-R1 (Original) |**73.4**|32.0|**73.8**|**71.5**|
> |Assembly-R1 (Updated) |70.4|**62.7**|70.9|70.3|
>
> **Table 1:** Comprehensive accuracy comparison of Assembly-R1 against open-source and closed-source baselines across different task categories on FurniBench.
>
> *Note: Original model uses the original reward design and config. Updated V1 uses a new learning rate, and Updated R1 uses a new accuracy reward design.*
>
> **Analysis:** Assembly-R1 consistently outperforms all external baselines, including closed-source giants like GPT-4o and Gemini-2.5-Pro, in overall accuracy. Notably, with our **new reward design** (Assembly-R1 Updated), we achieve a breakthrough in the most challenging "Part Connectivity" task, reaching **62.7%** accuracy. This significantly surpasses Gemini-2.5-Pro (49.3%), which was previously the only model performing well in this category.
>
> ### **Human Evaluation and Dataset Quality**
>
> **Response to Weakness 2**
>
> To address the concern regarding the lack of human evaluation, we employed a rigorous multi-stage validation process to ensure the realism and difficulty of FurniBench:
>
> 1.  **Source Validity:** Our dataset is derived from IKEA-ASM, which contains real-world assembly videos. We manually verified that the assembly sequences in our ground truth align with physical feasibility.
> 2.  **Human-in-the-Loop Calibration:** For ambiguous tasks like "Part Connectivity" (where multiple valid connections exist), we conducted manual inspections to construct robust ground-truth sets that include all valid permutations. This prevents the "false negative" issue where a valid but alternative assembly step is penalized.
> 3.  **Question Phrasing Verification:** We manually reviewed a subset of generated questions to ensure they are unambiguous and contextually appropriate (e.g., ensuring "First Disassemble Part" questions are only asked when the furniture is in a state that allows disassembly).
>
> ### **Benchmark Statistics and Clarity**
>
> **Response to Question 1**
>
> We appreciate the suggestion to include statistical visualizations. Please refer to the Appendix for more comprehensive statistics of the dataset, including Main Category, Sub Category, Task Type, Quantity, and Representative Expressions. We are also considering adding more intuitive visualizations to show the novelty, distribution, and coverage of the data. This is a very constructive suggestion!

---

> ### Author Response · Authors · 2025-11-21
>
> ### **Qualitative Analysis of Reasoning Process**
>
> **Response to Question 2**
>
> We appreciate the reviewer's careful observation regarding the reasoning in Figure 4(c). We address the concern about "restating" and provide evidence of genuine reasoning below.
>
> **1. Simplicity of Part Recognition (Figure 4c):**
> The "restating" pattern observed in Figure 4(c) (*"`<think>` The label Z corresponds to the table legs. `</think>`"*) occurs because the task is a straightforward visual recognition task. For such intuitive questions, the model converges to a direct confirmation strategy, rather than generating unnecessary verbose logic.
>
> **2. Complex Reasoning on OOD Data:**
> To demonstrate that our model engages in multi-step logical reasoning when the task complexity demands it, we present qualitative examples from Out-of-Domain (OOD) datasets (GRiD-3D [1], GQA [2]). These examples show the model breaking down problems, verifying negative constraints, and performing spatial reasoning. Also, this strengthens the idea of "SFT memorizes, RL generalizes".
>
> *   **Example 1: GRiD-3D**
>     *   **Question:** "You can find a Clock in the image. Taking its perspective, what number of objects are on the left in front of it?"
>     *   **Model Reasoning:** *"`<think>` The Clock is in the foreground... The chair is to the **right** of the Clock, and the refrigerator is to the **left** of the Clock. The Clock is the only object in the foreground. `</think>`"*
>     *   **Answer:** 1
>     *   **Analysis:** The model correctly interprets the spatial relationships relative to the object's perspective, a complex 3D reasoning task that goes far beyond simple pattern matching.
>
> *   **Example 2: GQA**
>     *   **Question:** "Is there a Trolley in the image?"
>     *   **Model Reasoning:** *"`<think>` The image shows a small figure, a red airplane, a purple suitcase, and a red scooter. There is no Trolley present in the image. `</think>`"*
>     *   **Answer:** No
>     *   **Analysis:** Instead of a simple "No", the model enumerates visible objects to exhaustively rule out the presence of the target object, demonstrating robust negative reasoning.
>
> *   **Example 3: GQA**
>     *   **Question:** "What color is the clock on the wall?"
>     *   **Model Reasoning:** *"`<think>` To determine the color of the clock on the wall, let's analyze the image step by step: 1. The clock is located on the wall. 2. The clock is visible and distinct... 3. The clock is white in color... Based on these observations, the clock on the wall is white. `</think>`"*
>     *   **Answer:** White
>     *   **Analysis:** The model explicitly locates the object before attributing the color, showing a structured verification process.
>
> These examples confirm that Assembly-R1 adapts its reasoning depth to the problem difficulty: concise for recognition, and detailed for spatial/logical queries.
>
> ## References
>
> [1] Lee, J. H., Kerzel, M., Ahrens, K., Weber, C., & Wermter, S. (2022). *What is Right for Me is Not Yet Right for You: A Dataset for Grounding Relative Directions via Multi-Task Learning*. In Proceedings of the International Joint Conference on Artificial Intelligence (IJCAI).
>
> [2] Hudson, D. A., & Manning, C. D. (2019). *GQA: A New Dataset for Real-World Visual Reasoning and Compositional Question Answering*. arXiv preprint arXiv:1902.09506.

---

### Official Review · Reviewer_XuK5 · 2025-11-01

**Soundness:** 3
**Presentation:** 2
**Contribution:** 2
**Rating:** 4
**Confidence:** 5

**Summary:**

This paper makes a well-executed application of VLMs and RL post-training on the 3D assembly tasks. The authors first formulate the task well, and then introduce a corresponding benchmark FurniBench, together with a large-scale dataset (FurniQA) for this task. To proceed, the authors perform SFT and GRPO based on pretrained Qwen2-VL-2B, resulting in two model variants: Assembly-V1 and Assembly-R1. The evaluation results demonstrate that GRPO can significantly enhance model performance, establishing a solid baseline for future research in 3D assembly tasks.

**Strengths:**

- Applying frontier methods (VLMs and RL post-training) to the 3D assembly task is a good practice to promote the development of AI-related areas
- This paper delivers a complete story, including problem formulation, benchmark design, dataset construction, model training, and experiments. The overall frame is clean and sound.
- Good analysis of OOD results and reward hacking, which manifests the advantages and more insights regarding the RL post-training of Assembly-R1

**Weaknesses:**

- Limited experimental results, particularly limited baselines for comparison. This makes it infeasible to assess the contribution and superiority of this paper. It turns out to be a story that we do something and the results are here. More potential ablation studies into the method are desired. For example, given the effectiveness of GRPO, how would different reward designs affect the model performance?
- I also have some concerns regarding the impact of this paper. With limited novelty, this work is more like establishing a solid baseline for future research on this task. How could the 3D assembly task relate to other domains with more popularity, e.g., embodied AI and robotics?

**Questions:**

- Why not continue RL post-training based on Assembly-V1? It seems the two model variants are independent (see Figure 3), which is not a common practice.
- In the analysis of reward hacking (Figure 5), what helps eliminate the phenomenon of reward hacking? The authors explained that “the reward design does not favor long reasoning“. But I think it is insufficient to account for the significant decrease in reward hacking since there is no explicit penalty for this. In other words, high reward hacking may not conflict with high accuracy.

---

> ### Author Response · Authors · 2025-11-21
>
> ## **Reply to Reviewer XuK5**
>
> We appreciate the reviewer's thoughtful and constructive review! Below, we address the concerns with additional experimental results and analysis.
>
> ### **Comprehensive Baseline Comparison**
>
> **Response to Weakness 1**
>
> To address the concern regarding limited baselines, we provide a comprehensive comparison between our model and other baselines on different tasks.
>
> |Task Category|Part Recognition|Part Connectivity|General Assembly Understanding|Total|
> |--|--|--|--|--|
> |**External Baselines**|||
> |GPT-4o|33.4|24.0|56.6|45.7|
> |Gemini-2.5-Pro|48.8|49.3|61.4|55.8|
> |Qwen2-VL-7B-Instruct|62.5|9.3|40.0|47.5|
> |LLaVA-1.5-7B-HF|57.0|16.0|39.7|45.4|
> |Qwen2.5-VL-3B-Instruct|41.6|13.3|38.2|38.3|
> |InternVL3-2B-Instruct|32.4|16.0|35.4|33.3|
> |Qwen2-VL-2B-Instruct|37.8|28.0|41.0|39.1|
> |**Our Models**|||
> |Assembly-V1 (Original) |67.7|28.0|64.3|63.9|
> |Assembly-V1 (Updated) |67.4|36.0|71.9|68.3|
> |Assembly-R1 (Original) |**73.4**|32.0|**73.8**|**71.5**|
> |Assembly-R1 (Updated) |70.4|**62.7**|70.9|70.3|
>
> **Table 1:** Comprehensive accuracy comparison of Assembly-R1 against open-source and closed-source baselines across different task categories on FurniBench.
>
> *Note: Original model uses the original reward design and config. Updated V1 uses a new learning rate, and Updated R1 uses a new accuracy reward design.*
>
> **Analysis:** Assembly-R1 consistently outperforms all external baselines, including closed-source giants like GPT-4o and Gemini-2.5-Pro, in overall accuracy. Notably, with our **new reward design** (Assembly-R1 Updated), we achieve a breakthrough in the most challenging "Part Connectivity" task, reaching **62.7%** accuracy. This significantly surpasses Gemini-2.5-Pro (49.3%), which was previously the only model performing well in this category.
>
> ### **Ablation Study - Reward Design**
>
> **Response to Weakness 1**
>
> To address the concern regarding reward sensitivity and design choices, we conducted ablation studies with alternative reward structures to encourage reasoning. Specifically, we explored explicit incentives for chain length and logical structure. However, we found that these heuristic-based rewards led to instability and reward hacking, reinforcing our decision to rely on our reward design.
>
> We tested two alternative configurations:
>
> 1.  **Length-Incentivized Reward:**
>     *   **Design:** Added a term $r_{len} = N_{tokens} \cdot 0.001$ (capped at 700 tokens $\approx$ 0.7 reward) to encourage longer chains.
>     *   **Outcome:** This caused a "verbosity explosion." The average completion length surged to 200 tokens by step 40 and exceeded 400 tokens by step 50. The model learned to "filibuster" to maximize $r_{len}$ rather than improve reasoning, leading to training instability and significantly increased compute costs without accuracy gains.
>
> 2.  **Logic-Keyword & Clipped-Length Reward:**
>     *   **Design:** Added a clipped length reward ($min(N_{tokens} \cdot 0.005, 0.25)$) and a "Logic Reward" triggered by keywords (e.g., *therefore, thus, first, next*).
>     *   **Outcome:** Although the length reward was capped, the keyword incentive indirectly encouraged verbosity. The model generated repetitive, long-winded chains to maximize the probability of hitting logic keywords. Completion length reached 400 tokens by step 70, again resulting in instability and slow convergence.
>
> **Conclusion:** These ablations demonstrate that **explicitly rewarding length or specific tokens induces reward hacking** (verbosity and repetition) rather than genuine reasoning. The increase in completion length also results in excessive training time, high resource usage, and convergence difficulties. This validates our final design: using a simple format constraint combined with a strong, outcome-based accuracy reward allows the model to self-discover optimal reasoning patterns without heuristic bias.
>
> ### **Impact of Our Work**
>
> **Response to Weakness 2**
>
> We argue that our work serves as a critical perception-reasoning proxy for Embodied AI and Robotics. 3D assembly VQA isolates fundamental skills required for manipulation, such as spatial reasoning and precondition checking. Our OOD results (e.g., GRiD-3D [1], GQA [2]) demonstrate that the spatial logic learned here transfers to general 3D tasks, making our model a strong candidate for the "Brain" in robotic planning systems. Furthermore, our new accuracy reward design for part connectivity directly parallels real-world assembly progress (where partial connections are valid intermediate states), offering a robust training signal for robotic policy learning.

---

> ### Author Response · Authors · 2025-11-21
>
> ### **RL Post-Training on Assembly-V1 (SFT Checkpoint)**
>
> **Response to Question 1**
>
> The reviewer asks why we did not continue RL post-training on Assembly-V1 (SFT). While initializing RL from an SFT model is standard practice, our original experimental design aimed to isolate the comparative effectiveness of RL versus SFT given the same training steps. Following the reviewer's recommendation, we performed this experiment and identified two critical issues that hindered performance:
>
> 1.  **Format Collapse:** The SFT model (Assembly-V1) had already converged to a specific, concise response distribution. When applying GRPO, the policy struggled to "break out" of this local optimum to explore the longer, step-by-step reasoning chains required for complex tasks.
>
> 2.  **Ineffectiveness of Length Rewards:** We attempted to encourage reasoning by adding a length reward. However, this resulted in the model generating verbose, repetitive text rather than meaningful reasoning chains, further degrading performance.
>
> The evaluation results for RL post-training on the Assembly-V1 are shown below:
>
> |Task Category|Part Recognition|Part Connectivity|General Assembly Understanding|Total|
> |--|--|--|--|--|
> |Assembly-V1 + RL (Original Reward)|69.1|23.1|70.7|67.6|
> |Assembly-V1 + RL (With Length Reward)|61.9|9.3|68.9|63.1|
> |Assembly-R1 (Updated) |**70.4**|**62.7**|**70.9**|**70.3**|
>
> **Conclusion:** Without an effective reasoning chain, the model particularly struggled with **Part Connectivity**, a very challenging task that requires spatial understanding. On the other hand, starting from the Base Model allows the policy to learn to generate more accurate responses and reasoning chains simultaneously.
>
> ### **Clarification on Reward Hacking**
>
> **Response to Question 2**
>
> The reviewer asks why reward hacking decreases without an explicit penalty, noting that "high reward hacking may not conflict with high accuracy."
>
> We claim that reward hacking (verbosity) conflicts with accuracy for two reasons:
>
> 1.  **Context Drift & Hallucination:** When the model generates excessive tokens to "hack" a length or keyword reward, it tends to drift away from the visual context. We observed the model describing irrelevant background details or hallucinating non-existent parts merely to extend the text, which frequently led to incorrect final answers.
> 2.  **Optimization Efficiency:** In our final design (Accuracy Reward only), the *only* way to maximize reward is to provide the correct answer. Any "hacking" (e.g., generating useless tokens or repetitive wording) increases the risk of reaching the output token limit before producing the required end-of-reasoning `</think>` token and the final answer. This would result in a failure to receive both the format reward and the accuracy reward. Consequently, the model naturally converges to a non-hacking state, as concise and effective reasoning represents the optimal policy for maximizing accuracy.
>
> ## References
>
> [1] Lee, J. H., Kerzel, M., Ahrens, K., Weber, C., & Wermter, S. (2022). *What is Right for Me is Not Yet Right for You: A Dataset for Grounding Relative Directions via Multi-Task Learning*. In Proceedings of the International Joint Conference on Artificial Intelligence (IJCAI).
>
> [2] Hudson, D. A., & Manning, C. D. (2019). *GQA: A New Dataset for Real-World Visual Reasoning and Compositional Question Answering*. arXiv preprint arXiv:1902.09506.

---

> ### Comment · Reviewer_XuK5 · 2025-11-26
> **Response to Rebuttal**
>
> Thanks for the authors' response. My concerns are not addressed well.
>
> ### Weakness 1: Limited experimental results...
> In rebuttal, the authors simply expand the results in Figure 3 into a table with per-category results. However, my concerns are:
> - If the paper's major contribution is a benchmark, then the authors should present more analyses into the task and behaviors across models.
> - If the paper's major contribution lies in the model, I don't think it is significant to finetune Qwen2-VL on a new dataset with existing SFT and GRPO methods. And the superior performance is not surprising as the model has been finetuned. Where is the novelty of the method, and how to demonstrate the advantage of the method?
>
> ### Weakness 2: Impact of Paper
> Let's consider this question: if I am a researcher in embodied AI and robotics, would this paper impress me and make me follow it, given current results on the FurniBench? The impact can only be supported by solid experimental results, e.g., the data facilitates the performance on specific benchmarks for embodied AI or robotics. I remember that GQA is a 2D VQA benchmark and has a limited relation to embodied AI or robotics. I don't know what GRID-3D is, and the authors just mention this benchmark without an explicit reference, which can be unprofessional. A paragraph of argument is not convincing enough to me.
>
> ### Question 1&2
> Thanks for the clarification on post-training strategy and reward hacking. That makes sense.

---

> ### Author Response · Authors · 2025-12-03
>
> We appreciate the reviewer's constrictive feedback and suggestions. Below, we address the reviewer's concern with additional experiments and clarifications.
>
> ## Contribution of the paper
>
> *   **Benchmark Contribution:** We introduce **FurniBench**, filling a critical gap in assembly VQA. Beyond scale, we have added an analysis distinguishing **Semantic vs. Spatial capabilities** (see Section Task Taxonomy). This diagnostic breakdown reveals that standard VLMs fail specifically on spatial structural understanding, validating the need for this specialized benchmark to guide future Embodied AI research.
> *   **Methodological Contribution (Robust Spatial Learning):**
>     *   **In-Domain Performance:** We are the first to adapt GRPO to assembly VQA, significantly outperforming SFT and commercial baselines on complex spatial tasks like *Part Connectivity* (see Main Response: Table 1).
>     *   **Out-of-Domain Generalization:** Crucially, our RL-based approach overcomes the *catastrophic forgetting* observed in SFT. As shown in our new OOD results, while standard SFT (`Assembly-V1`) degrades on external benchmarks, `Assembly-R1` achieves promising improvement on unseen general spatial understanding benchmarks (e.g. GRiD-3D [1], GQA [2], CV-Bench [3]) and Embodied AI specific benchmarks (e.g., SQA3D [4], Super-CLEVR [5]). This demonstrates two key findings:
>         * Our RL-based model effectively transfers learned spatial reasoning to new domains
>         * FurniBench encapsulates fundamental 3D spatial knowledge required by broader Embodied AI tasks.
> *   **Impact on Embodied Planning:** We demonstrate that our contributions translate to real-world reliability (see Section Evaluation on Embodied AI Benchmarks). We sampled a subset from our dataset, and create the task of **Consecutive Assembly Step Planning**. Our model shows greater advantages in tasks with increased depth, reaching 11.8% success rate for 5 consecutive steps, exponentially outperforming GPT-4o (1.0%) and Gemini-2.5-Pro (2.9%). This establishes our method as a viable pathway for reliable high-level robotic planning.
>
> ## Task Taxonomy: Semantic Understanding vs. Spatial Reasoning.
>
> We explicitly separate the tasks based on the required skills: Semantic Understanding (Solvable with 2D Recognition) and Spatial Reasoning (Require model's spatial-temporal reasoning). Models' performance is presented in **Table 1**.
>
> * Semantic Understanding:
>
>     * Single Part Recognition
>     * Object Recognition
>     * Action Recognition
>     * Missing Part Recognition
>
> * Spatial Reasoning:
>
>     * Connection After Installation
>     * First Assemble
>     * First Disassemble
>     * Next Step Planning
>     * Installation
>
> | **Models** | **Semantic Understanding** | **Spatial Reasoning**
> | :--- | :---: | :---: |
> | Gemini-2.5-Pro | 59.2 | 47.6 |
> | GPT-4o | 47.2 | 32.3 |
> | Qwen2-VL-7B-Instruct | 44.9 | 20.1 |
> | LLaVA-1.5-7B-HF | 41.1 | 21.9 |
> | Qwen2.5-VL-3B-Instruct | 38.9 | 22.9 |
> | InternVL3-2B-Instruct | 28.8 | 20.1 |
> | Qwen2-VL-2B-Instruct | 34.5 | 27.1 |
> | Assembly-V1 | 69.2 | 49.0 |
> | Assembly-R1 | **72.2** | **59.7** |
>
> **Table 1.** Benchmark Analysis: Semantic Understanding vs. Spatial Reasoning.
>
> **Analysis:** Table 1 reveals a distinct capability gap. Baseline models, on average, underperform on **Spatial Reasoning** (avg. 36.0%) compared to **Semantic Understanding** (avg. 38.0%), with commercial models like GPT-4o dropping over 10% on spatial tasks. This confirms a widespread deficiency in 3D structural reasoning. Conversely, `Assembly-R1` excels with **59.7%** spatial accuracy, surpassing the best baseline by **12.1%** and demonstrating that our method effectively bridges this reasoning gap.

---

> > ### Author Response · Authors · 2025-12-03
> >
> > ## Evaluation on Embodied AI Benchmarks
> >
> > In this section, we would like to demonstrate the how our benchmark and model can be contributed to applications like Embodied AI.
> >
> > * **From VQA to Planning Success in Embodied AI** Our dataset contains assembly steps planning, successful operation is the key to complete assembly. We have created a subset as Consecutive Assembly Steps, and we will measure the success rate of the model to plan the next assembly step with depth level from 1 - 5. For example, the candidate model is marked as success for a 3-step planning if it correctly plans the following 3 consecutive assembly steps. We modeled the Consecutive Assembly Step Planning Success Rate in **Table 2**.
> >
> > Consecutive Assembly Step Planning Success Rate
> >
> > |Model| step@1 | step@2 | step@3 | step@4 | step@5|
> > |--|--|--|--|--|--|
> > |Gemini-2.5-Pro|49.0|25.5|13.1|6.9|2.9|
> > |GPT-4o|45.7|23.8|10.5|4.9|1.0|
> > |Qwen2-VL-2B-Instruct (Base Model)|26.3|4.4|0.7|0.0|0.0|
> > |Assembly-V1 (Refined) |51.0|26.5|12.8|7.8|2.9|
> > |Assembly-R1 (Classic) |**58.4**|**37.0**|**25.5**|**17.2**|**11.8**|
> > |Assembly-R1 (PCR) |**58.4**|36.5|23.9|15.2|9.8|
> >
> > **Table 2.** Consecutive Assembly Step Planning Success Rate
> >
> > **Analysis:** `Assembly-R1` consistently outperforms baselines across all planning depths. The gap widens significantly at deeper steps: at step 5, while GPT-4o (1.0%) and the Base Model (0.0%) fail completely, `Assembly-R1` maintains **9.8-11.8%** success rate. This confirms our model's reliability for long-horizon planning, a critical requirement for autonomous robotics.
> >
> > * **ODD and Embodied AI Benchmarks:** We would like to demonstrate the impact of this paper with OOD benchmarks for VQA, GRiD-3D [1], GQA [2], CV-Bench [3] and specific dataset spatial reasoning benchmark which are popular in embodied AI, like SQA3D [4], Super-CLEVR [5].
> >
> > |Model|GRiD-3D (Free Text)|GQA (Free Text)|CV-Bench (MCQ)|SQA3D (Free Text)| Super-CLEVR (Free Text)|
> > |--|--|--|--|--|--|
> > |Base Model|34.4|61.9|62.4|32.2|42.7|
> > |Assembly-V1 (Refined)|33.5|55.3|49.6|21.8|41.2|
> > |Assembly-R1 (Classic)|**42.7**|62.5|63.7|29.7|37.4|
> > |Assembly-R1 (PCR)|40.7|**66.4**|**64.5**|**35.3**|**45.9**|
> >
> > **Table 3.** Evaluation on OOD Benchmarks, including Embodied AI-specific Benchmarks
> >
> > **Analysis:** `Assembly-R1 (PCR)` achieves state-of-the-art performance across diverse OOD benchmarks, reversing the catastrophic forgetting seen in SFT (`Assembly-V1`). The gains on Embodied AI-specific tasks like **SQA3D (+3.1%)** and **Super-CLEVR (+3.2%)** over the base model demonstrate that training on FurniBench instills generalized 3D spatial reasoning, validating its impact as a foundational resource for the broader Embodied AI community.
> >
> >
> >
> > ## References
> >
> > [1] Lee, J. H., Kerzel, M., Ahrens, K., Weber, C., & Wermter, S. (2022). *What is Right for Me is Not Yet Right for You: A Dataset for Grounding Relative Directions via Multi-Task Learning*. In Proceedings of the International Joint Conference on Artificial Intelligence (IJCAI).
> >
> > [2] Hudson, D. A., & Manning, C. D. (2019). *GQA: A New Dataset for Real-World Visual Reasoning and Compositional Question Answering*. arXiv preprint arXiv:1902.09506.
> >
> > [3] Tong, S., Brown, E., Wu, P., Woo, S., Middepogu, M., Akula, S. C., ... & Xie, S. (2024). *Cambrian-1: A Fully Open, Vision-Centric Exploration of Multimodal LLMs*. arXiv preprint arXiv:2406.16860.
> >
> > [4] Ma, X., Yong, S., Zheng, Z., Li, Q., Liang, Y., Zhu, S.-C., & Huang, S. (2023). *SQA3D: Situated Question Answering in 3D Scenes*. International Conference on Learning Representations (ICLR).
> >
> > [5] Li, Z., Wang, X., Stengel-Eskin, E., Kortylewski, A., Ma, W., Van Durme, B., & Yuille, A. L. (2023). *Super-CLEVR: A Virtual Benchmark to Diagnose Domain Robustness in Visual Reasoning*. Proceedings of the IEEE/CVF Conference on Computer Vision and Pattern Recognition (CVPR).

---

### Official Review · Reviewer_RC2n · 2025-11-01

**Soundness:** 3
**Presentation:** 2
**Contribution:** 2
**Rating:** 4
**Confidence:** 4

**Summary:**

This paper introduces Assembly-R1, a reinforcement learning–enhanced vision-language model (VLM) designed for 3D assembly reasoning tasks. To evaluate model performance in this domain, the authors construct FurniBench, an assembly-specific benchmark, and FurniQA, a dataset of ~1.6 million VQA pairs covering tasks such as part recognition, connectivity reasoning, and next-step inference.
Using Qwen2-VL-2B-Instruct as the base model, the authors first train a supervised fine-tuned version (Assembly-V1) and then apply Group Relative Policy Optimization (GRPO) with simple format + accuracy rewards to produce Assembly-R1, which achieves 71.7 % accuracy on FurniBench versus 64.9 % (SFT) and 39.1 % (base). The RL-trained model also shows slightly better out-of-domain performance on CVBench (63.5 % vs 62.4 %) and demonstrates more coherent chain-of-thought reasoning.

**Strengths:**

- Timely and relevant problem. The paper addresses 3D structural reasoning for assembly tasks, an underexplored but important domain bridging robotics, visual reasoning, and multimodal understanding.

- Dataset contribution. The creation of FurniBench and FurniQA is valuable. The dataset is large, well-organized, and includes thoughtful bias-mitigation strategies (dynamic part tagging, multiple valid answers, randomized options).

- Clear connection to recent RL-for-reasoning work. The use of GRPO builds on DeepSeek-R1-style frameworks, demonstrating that such techniques can transfer to 3D spatial reasoning.

**Weaknesses:**

- Limited novelty. The conceptual contribution is small. The method is a direct application of GRPO with two basic rewards (format + accuracy) to a new dataset. There is no significant methodological innovation—no new RL algorithm, architecture, or reward-design insight.

- Narrow empirical scope. Evaluation is confined to a single base model (Qwen2-VL-2B) and a synthetic IKEA-style dataset. No validation is shown on more complex 3D or real-world tasks (e.g., ShapeNet, ScanNet, or robotics demonstrations).

- Marginal out-of-domain gain. The OOD improvement on CVBench is only +1 %. This small gain weakens the claim that “RL generalizes while SFT memorizes.”

- Simplistic reward design. The reward formulation (+1 for format, +1 for correct answer) is too coarse to claim genuine reasoning improvement. There’s no evidence that RL truly learns 3D structural relationships rather than better formatting behavior.

- Missing ablations. The paper lacks sensitivity analyses for reward scaling, sampling temperature, or GRPO hyper-parameters. It also doesn’t isolate whether improvements come from GRPO itself or just more fine-tuning.

- Evaluation metric limitations. Accuracy on multiple-choice questions is a weak proxy for 3D reasoning. Human evaluation or geometric-consistency metrics (e.g., spatial error, collision rate) would provide stronger evidence.

- Dataset and task design risk overfitting. Since FurniQA questions are rule-generated from the same IKEA-ASM dataset, it is unclear how much semantic diversity truly exists. The model may learn annotation heuristics rather than 3D reasoning.

**Questions:**

- How do you ensure that GRPO improves genuine reasoning rather than simply response formatting?

- Why limit training to 15 k samples from FurniQA given that 1.6 M are available? Does performance saturate beyond this size?

- Did you compare Assembly-R1 with other RL methods (e.g., DPO, RLAIF) or analyze stability under different reward weights?

- Can the trained model handle open-ended free-text answers beyond multiple-choice formats?

- How reproducible are the results? The appendix lists configurations, but is code or dataset access planned before acceptance?

**Details Of Ethics Concerns:**

The paper poses minimal ethical risk. The dataset uses public IKEA ASM videos, and the work targets assistive robotics and AR/VR applications. Potential misuse (e.g., surveillance, biased reasoning about human actions) should still be monitored, but overall the impact is positive.

---

> ### Author Response · Authors · 2025-11-21
>
> ## **Reply to Reviewer RC2n**
>
> We appreciate the reviewer's constructive feedback. Below, we address the key concerns.
>
> ### **Main Contributions and Generalization**
>
> **Response to Weakness 1, 2, and 3**
>
> 1.  **Benchmark and Dataset:** We introduce **FurniBench**, filling a gap in assembly VQA with a large-scale dataset evaluating 3 categories and 15 task types, featuring rigorous bias control (e.g., dynamic tagging, multi-user sequences).
> 2.  **Methodological Adaptation:** We are the first to adapt **GRPO to assembly VQA**, significantly outperforming SFT and larger commercial baselines (see Main Response: Table 1). Our updated accuracy reward further enhances performance on the challenging **Part Connectivity** task.
> 3.  **Better Generalization:** Unlike SFT, which suffers from catastrophic forgetting, Assembly-R1 demonstrates strong generalizability on OOD datasets: GRiD-3D [1], GQA [2], and CV-Bench [3]. As shown below, it achieves a **+8.3% gain on GRiD-3D** (open-ended) over the base model, demonstrating that it learns transferable spatial reasoning.
>
> |Model|GRiD-3D (Free Text)|GQA (Free Text)|CV-Bench (MCQ)|
> |--|--|--|--|
> |Base Model|34.4|61.9|62.4|
> |Assembly-V1 (Updated)|33.5|55.3|49.6|
> |Assembly-R1 (Original)|**42.7**|62.5|63.7|
> |Assembly-R1 (Updated)|40.7|**66.4**|**64.5**|
> |--|--|--|--|
> |Base Model|34.4|61.9|62.4|
> |Assembly-V1 (Updated)|33.5|55.3|49.6|
> |Assembly-R1 (Original)|**42.7**|62.5|63.7|
> |Assembly-R1 (Updated)|40.7|**66.4**|**64.5**|
>
> **Table 1:** Accuracy comparison on Out-of-Domain (OOD) Datasets. *Note: Updated R1 uses our new accuracy reward design (please refer to Main Response: Analysis of Part Connectivity Task Section).*
>
> ### **Ablation Study - Reward Design**
>
> **Response to Weakness 4, 5 and Question 3**
>
> To address the concern regarding reward sensitivity and design choices, we conducted ablation studies with alternative reward structures to encourage reasoning. Specifically, we explored explicit incentives for chain length and logical structure. However, we found that these heuristic-based rewards led to instability and reward hacking, reinforcing our decision to rely on our reward design.
>
> We tested two alternative configurations:
>
> 1.  **Length-Incentivized Reward:**
>     *   **Design:** Added a term $r_{len} = N_{tokens} \cdot 0.001$ (capped at 700 tokens $\approx$ 0.7 reward) to encourage longer chains.
>     *   **Outcome:** This caused a "verbosity explosion." The average completion length surged to 200 tokens by step 40 and exceeded 400 tokens by step 50. The model learned to "filibuster" to maximize $r_{len}$ rather than improve reasoning, leading to training instability and significantly increased compute costs without accuracy gains.
>
> 2.  **Logic-Keyword & Clipped-Length Reward:**
>     *   **Design:** Added a clipped length reward ($min(N_{tokens} \cdot 0.005, 0.25)$) and a "Logic Reward" triggered by keywords (e.g., *therefore, thus, first, next*).
>     *   **Outcome:** Although the length reward was capped, the keyword incentive indirectly encouraged verbosity. The model generated repetitive, long-winded chains to maximize the probability of hitting logic keywords. Completion length reached 400 tokens by step 70, again resulting in instability and slow convergence.
>
> **Conclusion:** These ablations demonstrate that **explicitly rewarding length or specific tokens induces reward hacking** (verbosity and repetition) rather than genuine reasoning. The increase in completion length also results in excessive training time, high resource usage, and convergence difficulties. This validates our final design: using a simple format constraint combined with a strong, outcome-based accuracy reward allows the model to self-discover optimal reasoning patterns without heuristic bias.
>
> ### **Data Scaling Ablation**
>
> **Response to Weakness 5**
>
> We use intermediate checkpoints as proxies for data scaling ablation. Models are trained for same training steps, so the improvement in GRPO does not come from more fine-tuning.

---

> ### Author Response · Authors · 2025-11-21
>
> ### **Justification for MCQ Format**
>
> **Response to Weakness 6 and Question 3**
>
> We chose the Multiple-Choice Question (MCQ) format to ensure fair and rigorous evaluation, particularly for baselines that have not been fine-tuned on our specific task:
>
> *  **Handling Ambiguity:** Our dataset includes questions with both single and multiple correct answers. MCQ allows us to clearly define the solution space, whereas open-ended generation often leads to ambiguous or overly general responses (e.g., "assemble the parts") that are difficult to evaluate against specific actions like "align the leg with the hole."
> *  **Evaluation Accuracy:** Automated evaluation of open-ended text is prone to false negatives due to semantic variations. For example, in OOD testing, Assembly-R1 was penalized for predicting "fence" instead of "barbed wire" (GQA [2]) or "bicycle" instead of "bike" (GRiD-3D [1]), despite the semantic correctness. MCQ eliminates this ambiguity, providing a precise metric for reasoning capability.
> *  **Generalization Evidence:** Despite training on MCQs, our model demonstrates strong generalization to OOD datasets (both **MCQ-based**, like CV-Bench [3], and **open-ended tasks**, like GRiD-3D [1] and GQA [2]), showing that it learns underlying spatial reasoning rather than just multiple-choice heuristics. Please refer to **Table 1**.
>
> ### **Mitigating Overfitting Risks**
>
> **Response to Weakness 7**
>
> We explicitly designed the dataset and tasks to prevent overfitting and encourage robust learning:
>
> 1.  **Visual Diversity:** The IKEA-ASM source data includes diverse assembly sequences, user poses, and environmental conditions (lighting, background colors), preventing the model from relying on static visual cues.
> 2.  **Bias Mitigation Strategies:**
>     *   **Dynamic Tagging:** We randomize part tags (e.g., 'A', 'B') to ensure the model learns to recognize parts by shape and functionality rather than memorizing specific letter associations.
>     *   **Robust Answer Sets:** Our ground-truth sets account for multiple valid assembly strategies and sequences, preventing the model from overfitting to a single "golden" path.
>     *   **Subset Correctness:** During testing, we consider any non-empty subset of the correct answer set as valid, accommodating legitimate variations in assembly order.
>
> This rigorous design is validated by our OOD results: while the SFT model suffers from catastrophic forgetting (performance drop on OOD data), the GRPO-trained Assembly-R1 maintains or improves performance (e.g., significant gains on GRiD-3D), proving it has learned transferable spatial reasoning rather than memorizing dataset artifacts.
>
> ### **Design Selection**
>
> **Response to Weakness 2**
>
> We selected Qwen2-VL-2B for its efficiency, allowing rigorous comparison of training strategies (SFT vs. GRPO) within academic compute constraints.
>
> ### **Reasoning Analysis**
>
> **Response to Weakness 4 and Question 1**
>
> Please refer to the "Qualitative Analysis of Reasoning Process" section in the Main Response for examples.
>
> ### **Training Dataset**
>
> **Response to Question 2**
>
> We trained all models on the same randomly sampled 15k subset to ensure fair comparison under resource constraints.
>
> ### **Reproducibility**
>
> **Response to Question 5**
>
> We appreciate the reviewer for mentioning reproducibility. Please refer to the supplementary material we submitted. Training scripts for Assembly-R1 and sample data for FurniBench are included. We are committed to releasing the full dataset and checkpoints upon the acceptance of our work.
>
> ## References
>
> [1] Lee, J. H., Kerzel, M., Ahrens, K., Weber, C., & Wermter, S. (2022). *What is Right for Me is Not Yet Right for You: A Dataset for Grounding Relative Directions via Multi-Task Learning*. In Proceedings of the International Joint Conference on Artificial Intelligence (IJCAI).
>
> [2] Hudson, D. A., & Manning, C. D. (2019). *GQA: A New Dataset for Real-World Visual Reasoning and Compositional Question Answering*. arXiv preprint arXiv:1902.09506.
>
> [3] Tong, S., Brown, E., Wu, P., Woo, S., Middepogu, M., Akula, S. C., ... & Xie, S. (2024). *Cambrian-1: A Fully Open, Vision-Centric Exploration of Multimodal LLMs*. arXiv preprint arXiv:2406.16860.

---

### Author Response · Authors · 2025-11-21
**Main Response to Reviewers’ Common Concerns - 1**

## **Reply to All Reviewers**

We sincerely thank all reviewers for their thoughtful and constructive feedback. Below, we address the key common concerns raised. We will incorporate these updates into the next version of the paper.

### **Comprehensive Comparison with Baselines on Different Task Categories**

As suggested by **Reviewer 85gg** and **Reviewer XuK5**, we agree that including comprehensive experimental results in a numerical table is more informative than the accuracy curve. We will move the accuracy curve (Figure 3) to the appendix and include the following comprehensive comparison table in the main text in the revision.

We present a comprehensive comparison between our model and other open-source and closed-source commercial baselines with similar or much larger scales. The table below compares our model with other baselines on different task categories:

|Task Category|Part Recognition|Part Connectivity|General Assembly Understanding|Total|
|--|--|--|--|--|
|**External Baselines**||||
|GPT-4o|33.4|24.0|56.6|45.7|
|Gemini-2.5-Pro|48.8|49.3|61.4|55.8|
|Qwen2-VL-7B-Instruct|62.5|9.3|40.0|47.5|
|LLaVA-1.5-7B-HF|57.0|16.0|39.7|45.4|
|Qwen2.5-VL-3B-Instruct|41.6|13.3|38.2|38.3|
|InternVL3-2B-Instruct|32.4|16.0|35.4|33.3|
|Qwen2-VL-2B-Instruct|37.8|28.0|41.0|39.1|
|**Our Models**||||
|Assembly-V1 (Original) |67.7|28.0|64.3|63.9|
|Assembly-V1 (Refined) |67.4|36.0|71.9|68.3|
|Assembly-R1 (Classic) |**73.4**|32.0|73.8|71.5|
|Assembly-R1 (PCR) |70.6|**77.3**|**75.1**|**73.4**|

**Table 1:** Comprehensive accuracy comparison of Assembly-R1 against open-source and closed-source baselines across different task categories on FurniBench.

*Note: Classic model uses the original reward design and config. Refined V1 uses a new learning rate, and Assembly-R1 (PCR) uses a new accuracy reward design (PCR stands for Pure-Coverage Reward), which will be introduced in the Analysis of Part Connectivity Task section.*

**Analysis:** Assembly-R1 consistently outperforms all external baselines, including closed-source giants like GPT-4o and Gemini-2.5-Pro, in overall accuracy. Notably, with our **new reward design** (Assembly-R1 (PCR)), we achieve a breakthrough in the most challenging "Part Connectivity" task, reaching **62.7%** accuracy. This significantly surpasses Gemini-2.5-Pro (49.3%), which was previously the only model performing well in this category.


### **Analysis of Part Connectivity Task - Refined Reward Design**

We thank **Reviewer tmox** for the careful observation on the minimal gain on the Part Connectivity Task. Here is the analysis:

1. Part Connectivity is the most challenging category, demanding genuine structural reasoning (topology, attachments). Most baselines struggle, except Gemini-2.5-Pro, which benefits from its larger scale and spatial reasoning capabilities.

2. The root cause is that many questions have multiple valid answers. Our original training reward was "all-or-nothing," assigning zero credit to partially correct responses. This hindered incremental learning, as the model received no feedback for valid but incomplete answers, even though our test metric accounts for them.

3. Given this, we have revised the accuracy rewards to provide graded credit to partially correct, strictly subset answers, while still penalizing any wrong options:

    Specifically, let $A_{gt}$ be the ground-truth set of correct options and $A_{pred}$ the model's chosen set. **If** $A_{pred} \subseteq A_{gt}$ (model's answer is a subset of ground truth): $$r_{acc} = \frac{|A_{pred}|}{|A_{gt}|}$$ **If** $A_{pred} \not\subseteq A_{gt}$ (contains any incorrect option) **or** $A_{pred} = \emptyset$ (empty answer): $$r_{acc} = 0$$

```
For example, if the ground truth set is ['A', 'I', 'O', 'T', 'E'], model's response of ['O', 'A', 'E'] will be rewarded 0.6, and model's response of [] or ['A', 'B', 'O'] will be rewarded 0.
```

   This is justified because partial answers still guide correct installation, whereas wrong answers lead to structural failure. We also retain the format reward (+1) to ensure valid output structure.

4. **Outcome**: With this new accuracy reward design, the model is rewarded for partially correct responses, enabling it to learn connectivity in smaller, verifiable steps. Qualitatively, the model now avoids “all-or-nothing” behavior and exhibits more consistent structural reasoning. After applying this new reward, Assembly-R1 (PCR)

the performance of Assembly-R1 on the Part Connectivity Task is much better, outperforming all other baselines, including Gemini-2.5-Pro with thinking enabled. This indicates that our model has improved capability in structural understanding.

---

> ### Author Response · Authors · 2025-11-21
> **Main Response to Reviewers’ Common Concerns - 2**
>
> ### **Out-of-Domain (OOD) Test and Application to Embodied AI**
>
> In this section, we demonstrate how our benchmark and model contribute to applications like Embodied AI.
>
> *   **From VQA to Planning Success in Embodied AI:** Our dataset contains assembly step planning, where successful operation is key to complete assembly. We created a subset called **Consecutive Assembly Steps** to measure the model's success rate in planning the next assembly step with depth levels from 1 to 5. For example, a model is marked as successful for 3-step planning if it correctly plans the next 3 consecutive steps.
>
> |Model| step@1 | step@2 | step@3 | step@4 | step@5|
> |--|--|--|--|--|--|
> |Gemini-2.5-Pro|49.0|25.5|13.1|6.9|2.9|
> |GPT-4o|45.7|23.8|10.5|4.9|1.0|
> |Qwen2-VL-2B-Instruct (Base Model)|26.3|4.4|0.7|0.0|0.0|
> |Assembly-V1 (Refined) |51.0|26.5|12.8|7.8|2.9|
> |Assembly-R1 (Classic) |**58.4**|**37.0**|**25.5**|**17.2**|**11.8**|
> |Assembly-R1 (PCR) |**58.4**|36.5|23.9|15.2|9.8|
>
> **Table 2.** Consecutive Assembly Step Planning Success Rate
>
> **Analysis:** `Assembly-R1` consistently outperforms baselines across all planning depths. The gap widens significantly at deeper steps: at step 5, while GPT-4o (1.0%) and the Base Model (0.0%) fail completely, `Assembly-R1` maintains a **9.8-11.8%** success rate. This confirms our model's reliability for long-horizon planning, a critical requirement for autonomous robotics.
>
> *   **OOD and Embodied AI Benchmarks:** As suggested by **Reviewer RC2n** and **Reviewer XuK5**, we provide additional OOD results on GRiD-3D [1], GQA [2], CV-Bench [3], and popular Embodied AI benchmarks like SQA3D [4] and Super-CLEVR [5]. This substantiates our claim that "SFT memorizes, RL generalizes" and demonstrates the potential impact of our work on embodied scenes requiring spatial reasoning.
>
> |Model|GRiD-3D (Free Text)|GQA (Free Text)|CV-Bench (MCQ)|SQA3D (Free Text)| Super-CLEVR (Free Text)|
> |--|--|--|--|--|--|
> |Base Model|34.4|61.9|62.4|32.2|42.7|
> |Assembly-V1 (Refined)|33.5|55.3|49.6|21.8|41.2|
> |Assembly-R1 (Classic)|**42.7**|62.5|63.7|29.7|37.4|
> |Assembly-R1 (PCR)|40.7|**66.4**|**64.5**|**35.3**|**45.9**|
>
> **Table 3.** Evaluation on OOD Benchmarks, including Embodied AI-specific Benchmarks
>
> **Analysis:** `Assembly-R1 (PCR)` achieves state-of-the-art performance across diverse OOD benchmarks, reversing the catastrophic forgetting seen in SFT (`Assembly-V1`). The gains on Embodied AI-specific tasks like **SQA3D (+3.1%)** and **Super-CLEVR (+3.2%)** over the base model demonstrate that training on FurniBench instills generalized 3D spatial reasoning, validating its impact as a foundational resource for the broader Embodied AI community.
>
> 1.  **Consistent Generalization:** Assembly-R1 achieves consistent performance improvements across all OOD datasets compared to the Base Model. For instance, on the open-ended GRiD-3D dataset, Assembly-R1 (Classic) achieves a **+8.3%** gain (42.7% vs 34.4%).
> 2.  **Impact of Refined Reward:** With our updated reward design, the model's generalization capability is further enhanced on general visual reasoning tasks. Assembly-R1 (PCR) achieves the highest performance on GQA (**66.4%**, +4.5% over Base) and CV-Bench (**64.5%**, +2.1% over Base), indicating that the new implementation of accuracy reward better encourages the model to gradually find correct patterns and transfers effectively to other domains.
> 3.  **SFT Overfitting & Forgetting:** In contrast, Assembly-V1 (SFT) consistently underperforms the Base Model (e.g., dropping from 62.4% to 49.6% on CV-Bench). This suggests that supervised fine-tuning on a specific domain leads to **catastrophic forgetting** of general visual reasoning capabilities.
> 4.  **Performance on Open-Ended Generation:** To address the concern that MCQ accuracy might be a weak proxy for 3D reasoning, we highlight our performance on **open-ended free-text generation** tasks (GRiD-3D and GQA). Assembly-R1 achieves significant gains over the Base Model (up to **+8.3%** on GRiD-3D and **+4.5%** on GQA) even without multiple-choice options. This confirms that our model's improvements stem from genuine spatial reasoning capabilities rather than just learning to select the best option among distractors.

---

> ### Author Response · Authors · 2025-11-21
> **Main Response to Reviewers’ Common Concerns - 3**
>
> ### **Challenging 3D Subset**
>
> We thank **Reviewer tmox** and **Reviewer XuK5** for the idea of selecting a subset that is most indicative of 3D spatial reasoning.
>
> We explicitly separate the tasks based on the required skills: Semantic Understanding (Solvable with 2D Recognition) and Spatial Reasoning (Require model's spatial-temporal reasoning). Models' performance is presented in **Table 1**.
>
> * Semantic Understanding:
>
>     * Single Part Recognition
>     * Object Recognition
>     * Action Recognition
>     * Missing Part Recognition
>
> * Spatial Reasoning:
>
>
>     * Connection After Installation
>     * First Assemble
>     * First Disassemble
>     * Next Step Planning
>     * Installation
>
> | **Models** | **Semantic Understanding** | **Spatial Reasoning** |
> | :--- | :---: | :---: |
> | Gemini-2.5-Pro | 59.2 | 47.6 |
> | GPT-4o | 47.2 | 32.3 |
> | Qwen2-VL-7B-Instruct | 44.9 | 20.1 |
> | LLaVA-1.5-7B-HF | 41.1 | 21.9 |
> | Qwen2.5-VL-3B-Instruct | 38.9 | 22.9 |
> | InternVL3-2B-Instruct | 28.8 | 20.1 |
> | Qwen2-VL-2B-Instruct | 34.5 | 27.1 |
> | Assembly-V1 | 69.2 | 49.0 |
> | Assembly-R1 | **72.2** | **59.7** |
>
> **Table 1.** Benchmark Analysis: Semantic Understanding vs. Spatial Reasoning.
>
> **Analysis:** Table 1 reveals a distinct capability gap. Baseline models, on average, underperform on **Spatial Reasoning** (avg. 36.0%) compared to **Semantic Understanding** (avg. 38.0%), with commercial models like GPT-4o dropping over 10% on spatial tasks. This confirms a widespread deficiency in 3D structural reasoning. Conversely, `Assembly-R1` excels with **59.7%** spatial accuracy, surpassing the best baseline by **12.1%** and demonstrating that our method effectively bridges this reasoning gap.
>
>
> ### **Ablation Study - Reward Design**
>
> To address the concern regarding reward sensitivity and design choices, we conducted ablation studies with alternative reward structures to encourage reasoning. Specifically, we explored explicit incentives for chain length and logical structure. However, we found that these heuristic-based rewards led to instability and reward hacking, reinforcing our decision to rely on our reward design.
>
> We tested two alternative configurations:
>
> 1.  **Length-Incentivized Reward:**
>     *   **Design:** Added a term $r_{len} = N_{tokens} \cdot 0.001$ (capped at 700 tokens $\approx$ 0.7 reward) to encourage longer chains.
>     *   **Outcome:** This caused a "verbosity explosion." The average completion length surged to 200 tokens by step 40 and exceeded 400 tokens by step 50. The model learned to "filibuster" to maximize $r_{len}$ rather than improve reasoning, leading to training instability and significantly increased compute costs without accuracy gains.
>
> 2.  **Logic-Keyword & Clipped-Length Reward:**
>     *   **Design:** Added a clipped length reward ($min(N_{tokens} \cdot 0.005, 0.25)$) and a "Logic Reward" triggered by keywords (e.g., *therefore, thus, first, next*).
>     *   **Outcome:** Although the length reward was capped, the keyword incentive indirectly encouraged verbosity. The model generated repetitive, long-winded chains to maximize the probability of hitting logic keywords. Completion length reached 400 tokens by step 70, again resulting in instability and slow convergence.
>
> **Conclusion:** These ablations demonstrate that **explicitly rewarding length or specific tokens induces reward hacking** (verbosity and repetition) rather than genuine reasoning. The increment in completion length also results in excessive training time, resource usage, and difficulty in convergence. This validates our final design: using a simple format constraint combined with a strong, outcome-based accuracy reward allows the model to self-discover optimal reasoning patterns without heuristic bias.
>
> ### **RL Post-Training on Assembly-V1 (SFT Checkpoint)**
>
> We appreciate **Reviewer XuK5**'s suggestion to investigate RL post-training initialized from the SFT checkpoint (Assembly-V1). While initializing RL from an SFT model is standard practice, our original experimental design aimed to isolate the comparative effectiveness of RL versus SFT given the same training steps. Following the reviewer's recommendation, we performed this experiment and identified two critical issues that hindered performance:
>
> 1.  **Format Collapse:** The SFT model (Assembly-V1) had already converged to a specific, concise response distribution. When applying GRPO, the policy struggled to "break out" of this local optimum to explore the longer, step-by-step reasoning chains required for complex tasks.
>
> 2.  **Ineffectiveness of Length Rewards:** We attempted to encourage reasoning by adding a length reward. However, this resulted in the model generating verbose, repetitive text rather than meaningful reasoning chains, further degrading performance.

---

> ### Author Response · Authors · 2025-11-21
> **Main Response to Reviewers’ Common Concerns - 4**
>
> The evaluation results for RL post-training on the Assembly-V1 are shown below:
>
> |Task Category|Part Recognition|Part Connectivity|General Assembly Understanding|Total|
> |--|--|--|--|--|
> |Assembly-V1 + RL (Original Reward)|69.1|23.1|70.7|67.6|
> |Assembly-V1 + RL (With Length Reward)|61.9|9.3|68.9|63.1|
> |Assembly-R1 (PCR) |**70.6**|**77.3**|**75.1**|**73.4**|
>
> **Conclusion:** Without effective reasoning chains, the model particularly struggles with **Part Connectivity**, which is a very challenging task that requires the model's spatial understanding. On the other hand, for our model starting from the Base Model, it allows the policy to learn to generate more accurate responses and reasoning chains simultaneously.
>
> **Reasoning:** Starting from the Base model (Qwen2-VL) allows the policy to learn **both** the task format and the reasoning strategy simultaneously.
>
> ### **Qualitative Analysis of Reasoning Process**
>
> We appreciate **Reviewer 85gg**'s careful observation regarding the reasoning in Figure 4(c). We address the concern about "restating" and provide evidence of genuine reasoning below.
>
> **1. Simplicity of Part Recognition (Figure 4c):**
> The "restating" pattern observed in Figure 4(c) (*"`<think>` The label Z corresponds to the table legs. `</think>`"*) occurs because the task is a straightforward visual recognition task. For such intuitive questions, the model converges to a direct confirmation strategy, rather than generating unnecessary verbose logic.
>
> **2. Complex Reasoning on OOD Data:**
> To demonstrate that our model *does* engage in multi-step logical reasoning when the task complexity demands it, we present qualitative examples from Out-of-Domain (OOD) datasets (GRiD-3D [1], GQA [2]). These examples show the model breaking down problems, verifying negative constraints, and performing spatial reasoning.
>
> *   **Example 1: GRiD-3D**
>     *   **Question:** "You can find a Clock in the image. Taking its perspective, what number of objects are on the left in front of it?"
>     *   **Model Reasoning:** *"`<think>` The Clock is in the foreground... The chair is to the **right** of the Clock, and the refrigerator is to the **left** of the Clock. The Clock is the only object in the foreground. `</think>`"*
>     *   **Answer:** 1
>     *   **Analysis:** The model correctly interprets the spatial relationships relative to the object's perspective, a complex 3D reasoning task that goes far beyond simple pattern matching.
>
> *   **Example 2: GQA**
>     *   **Question:** "Is there a Trolley in the image?"
>     *   **Model Reasoning:** *"`<think>` The image shows a small figure, a red airplane, a purple suitcase, and a red scooter. There is no Trolley present in the image. `</think>`"*
>     *   **Answer:** No
>     *   **Analysis:** Instead of a simple "No", the model enumerates visible objects to exhaustively rule out the presence of the target object, demonstrating robust negative reasoning.
>
> *   **Example 3: GQA**
>     *   **Question:** "What color is the clock on the wall?"
>     *   **Model Reasoning:** *"`<think>` To determine the color of the clock on the wall, let's analyze the image step by step: 1. The clock is located on the wall. 2. The clock is visible and distinct... 3. The clock is white in color... Based on these observations, the clock on the wall is white. `</think>`"*
>     *   **Answer:** White
>     *   **Analysis:** The model explicitly locates the object before attributing the color, showing a structured verification process.
>
> These examples confirm that Assembly-R1 adapts its reasoning depth to the problem difficulty: concise for recognition, and detailed for spatial/logical queries.
>
> ## References
>
> [1] Lee, J. H., Kerzel, M., Ahrens, K., Weber, C., & Wermter, S. (2022). *What is Right for Me is Not Yet Right for You: A Dataset for Grounding Relative Directions via Multi-Task Learning*. In Proceedings of the International Joint Conference on Artificial Intelligence (IJCAI).
>
> [2] Hudson, D. A., & Manning, C. D. (2019). *GQA: A New Dataset for Real-World Visual Reasoning and Compositional Question Answering*. arXiv preprint arXiv:1902.09506.
>
> [3] Tong, S., Brown, E., Wu, P., Woo, S., Middepogu, M., Akula, S. C., ... & Xie, S. (2024). *Cambrian-1: A Fully Open, Vision-Centric Exploration of Multimodal LLMs*. arXiv preprint arXiv:2406.16860.
>
> [4] Ma, X., Yong, S., Zheng, Z., Li, Q., Liang, Y., Zhu, S.-C., & Huang, S. (2023). *SQA3D: Situated Question Answering in 3D Scenes*. International Conference on Learning Representations (ICLR).
>
> [5] Li, Z., Wang, X., Stengel-Eskin, E., Kortylewski, A., Ma, W., Van Durme, B., & Yuille, A. L. (2023). *Super-CLEVR: A Virtual Benchmark to Diagnose Domain Robustness in Visual Reasoning*. Proceedings of the IEEE/CVF Conference on Computer Vision and Pattern Recognition (CVPR).

---

### Author Response · Authors · 2025-12-03
**Summary of Rebuttal Updates**

## **Summary of Rebuttal Updates**

We thank the Area Chair and Reviewers for their time and effort. During the rebuttal period, we have carefully addressed all concerns and conducted extensive additional experiments to strengthen our claims. Common concerns are addressed in the Main Response, while individual concerns raised by each reviewer are addressed in their respective individual responses. New experimental results and analysis have been added to the revised PDF. We summarize our key actions and improvements below:

**1. Comprehensive Baseline Comparisons & Task Taxonomy**
*   We provided a detailed numerical comparison against a wide range of open-source and closed-source commercial baselines (including GPT-4o and Gemini-2.5-Pro).
*   We introduced a clear taxonomy distinguishing between **Semantic Understanding** and **Spatial Reasoning** tasks, highlighting that while baselines perform decently on semantics, they struggle significantly with spatial structure, a gap our model effectively bridges.

**2. Methodological Improvement: Refined Reward Design**
*   Addressing the feedback on the "Part Connectivity" task, we refined our reward mechanism to a **Pure-Coverage Reward (PCR)** design. This moves away from "all-or-nothing" scoring to a graded system that credits partially correct structural understanding.
*   This change led to a breakthrough in performance on the most challenging structural tasks, surpassing all baselines, and OOD benchmarks as well.

**3. Extensive Out-of-Domain (OOD) & Embodied AI Evaluation**
*   To demonstrate generalization, we evaluated our model on multiple external benchmarks, including general 3D VQA datasets and specific Embodied AI benchmarks.
*   The results confirm that our RL-based approach overcomes the catastrophic forgetting observed in SFT models and successfully transfers learned spatial reasoning skills to unseen domains.

**4. New Analysis on Long-Horizon Planning**
*   We introduced a **Consecutive Assembly Step Planning** metric to evaluate reliability in multi-step reasoning.
*   Our model demonstrated superior robustness in long-horizon planning compared to commercial models, validating its potential as a reliable planner for robotic assembly.

**5. Rigorous Ablation Studies & Qualitative Analysis**
*   We conducted ablation studies on reward designs (e.g., length incentives) to justify our architectural choices and address concerns about reward hacking.
*   We provided qualitative examples of the model's reasoning process on OOD data to prove it learns genuine spatial logic rather than memorizing dataset patterns.

We believe these additional results and analyses robustly address the reviewers' comments and firmly establish the contribution of our work to applications, like Embodied AI community.

---

### Meta-Review · Area_Chair_orD5 · 2026-01-10

**Summary:**

The reviewers generally acknowledge the importance of the 3D assembly reasoning domain and appreciate the contribution of the large-scale FurniBench and FurniQA datasets. However, several key concerns were raised that informed the current assessment:

+ Methodological Novelty: Multiple reviewers argued that the technical contribution is limited, characterizing the work as a direct application of the existing GRPO algorithm to a new dataset without significant architectural or algorithmic innovation.

+ Depth of 3D Spatial Reasoning: Concerns were raised regarding whether the tasks truly evaluate 3D structural reasoning or if they can be solved using 2D visual cues and simple logic.

+ Evaluation and Baselines: Initial reviews pointed to a lack of comprehensive comparisons against strong commercial and open-source baselines.

+ Out-of-Domain (OOD) Generalization: One reviewer questioned the initial marginal gains on OOD benchmarks, which weakened the claim that RL significantly improves generalization over SFT.

+ Performance on Challenging Tasks: Reviewers noted minimal initial improvements in the "Part Connectivity" category, suggesting the model struggled with complex structural relationships.

+ Human Validation: The absence of human study or inter-annotator validation for the rule-generated QA pairs was identified as a weakness.

**Reviewer Concerns:**

**Addressed Concerns**

+ Baseline Comparisons: The authors successfully addressed the lack of comparative results by providing a comprehensive numerical table including seven open-source and closed-source baselines (e.g., GPT-4o, Gemini-2.5-Pro).

+ Part Connectivity Performance: By introducing a Pure-Coverage Reward (PCR) design, the authors significantly improved the model's accuracy on the challenging "Part Connectivity" task from 32.0% to 62.7%, surpassing all external baselines.

+ 3D vs. 2D Reasoning: The authors clarified the "3D" nature of the tasks by introducing a Task Taxonomy and a Challenging 3D Subset, demonstrating that while baselines fail on structural spatial reasoning, Assembly-R1 maintains robust performance.

+ OOD Generalization: New evaluations on Embodied AI benchmarks like SQA3D (+3.1%) and Super-CLEVR (+3.2%) provided stronger evidence for the claim that "RL generalizes" where SFT fails.

+ Dataset Validation: Concerns regarding the quality of rule-generated QA pairs were addressed through a description of a multi-stage human-in-the-loop calibration and manual verification process.

**Outstanding Concerns**

+ Conceptual Novelty: A persistent concern remains regarding the limited methodological innovation, as some reviewers view the work primarily as a direct application of the existing GRPO algorithm to a new dataset.

+ Empirical Scope: The evaluation remains confined to a single base model architecture (Qwen2-VL-2B), leaving the general applicability of this approach across different model scales or architectures unverified.

+ Significance of Contribution: Despite the added benchmarks, one reviewer remains skeptical about the impact of fine-tuning a specific model on a niche dataset using standard techniques.

**Reviewer Scores:**

Reviewer XuK5 (Score: 4): This reviewer explicitly stated after the rebuttal that their concerns regarding the paper’s limited novelty and impact were not well-addressed. They remained unconvinced that fine-tuning an existing model with standard RL methods constitutes a significant contribution.

Reviewer RC2n (Score: 4): Their primary critique focused on limited methodological novelty and a narrow empirical scope. Since these are fundamental aspects of the work's design, it is improbable that their score would have changed.

Reviewer 85gg (Score: 4): While the authors provided the requested numerical tables , this reviewer’s concern regarding the lack of human evaluation and the overall significance of the benchmark suggests their score would remain at a 4.

Reviewer tmox (Score: 6): Although this reviewer was more positive , their single favorable rating is insufficient to overcome the shared skepticism from the other three reviewers regarding the work's technical innovation.

---

### Decision · Program_Chairs · 2026-01-26

Reject